

# Relative impact of aerosol, soil moisture and orography perturbations on deep convection

Linda Schneider, Christian Barthlott, and Corinna Hoose

Institute of Meteorology and Climate Research (IMK-TRO), Karlsruhe Institute of Technology (KIT), Germany

**Correspondence:** Christian Barthlott (christian.barthlott@kit.edu)

**Abstract.** The predictability of deep moist convection depends on many factors such as the synoptic-scale flow, the geographical region (i.e., the presence of mountains), land surface–atmosphere as well as aerosol–cloud interactions. This study addresses all these factors by investigating the relative impact of orography, soil moisture and aerosols on precipitation over Germany in different weather regimes. To this end, we conduct numerical sensitivity studies with the COnsortium for Small-sale MOdelling (COSMO) model at high spatial resolution (500 m grid spacing) for six days with weak and strong synoptic forcing. The numerical experiments consist of (i) successive smoothing of topographical features, (ii) systematic changes in the initial soil moisture fields (spatially homogeneous increase/decrease, horizontal uniform soil moisture, different realizations of dry/wet patches), and (iii) different assumptions on the ambient aerosol concentration (spatially homogeneous and heterogeneous fields). Our results show that the impact of these perturbations on precipitation is on average higher for weak than for strong synoptic forcing. Soil moisture and aerosols are each responsible for the maximum precipitation response for three of the cases, while the sensitivity to terrain forcing always shows the smallest spread. For the majority of the analyzed cases, the model produces a positive soil moisture–precipitation feedback when averaged over the entire model domain. Furthermore, correct initial values are much more important than the spatial distribution of dry and wet patches. The precipitation response to changes in the CCN concentration is more complex and case dependent. The smoothing of terrain shows weaker impacts on days with strong synoptic forcing because surface fluxes are less important and orographic ascent is still simulated reasonably well, despite missing fine-scale orographical features. We apply an object-based characterization to identify if and how the perturbations affect the structure, location, timing, and intensity of precipitation. These diagnostics reveal that the structure component is highest in the soil moisture and aerosol simulations, often due to changes in the maximum precipitation amounts. This indicates that the dominant mechanisms for convection initiation remain, but that precipitation amounts depend on the strength of the trigger mechanisms. Location and amplitude parameters are both much smaller. Still, the temporal evolution of the amplitude component correlates well with the rain rate. Our results suggest that for quantitative precipitation forecasting, both aerosols and soil moisture are of similar importance and that their inclusion in convective-scale ensemble forecasting containing classical sources of uncertainty should be assessed in the future.



## 1 Introduction

Forecasting convective precipitation remains one of the key challenges in numerical weather prediction (NWP) and has large social, economic and environmental impacts due to the multiple risks from hail, lightning, strong winds, and heavy precipitation. Convective precipitation results from a chain of complex processes and multi-scale interactions in the atmosphere and

is therefore accompanied by numerous uncertainties in its formation. Although convection-permitting models have provided a step-change in rainfall forecasting capabilities (Clark et al., 2016), current state-of-the art models still exhibit persistent and systematic shortcomings due to an inadequate representation of unresolved processes (Berner et al., 2017). This makes it difficult to properly predict convective precipitation, resulting in an often inadequate accuracy for many applications (Kühnlein et al., 2014; Mittermaier, 2014). The predictability of convective precipitation, i.e., the degree to which a correct prediction of

the state of the atmospheric can be made, depends on many aspects such as, among others, the synoptic-scale flow, the geographical region (i.e., the presence of mountains), the underlying land surface and microphysical uncertainties. For thermally forced convection, physical understanding is further challenged by the essential nonlinearity of thermally driven circulations, large spatial heterogeneity in thermodynamics and winds over complex terrain, and multi-scale interactions between the land surface and the planetary boundary layer (Kirshbaum et al., 2018). Land surface properties (e.g., land cover, terrain, and soil

texture) are highly heterogeneous across a wide range of spatiotemporal scales (Santanello et al., 2018) and potential linkages between land surface variables and atmospheric variables such as temperature and precipitation are difficult to establish (e.g., Seneviratne et al., 2010). Over mountainous terrain, thermally induced wind systems and low-level convergence zones are crucial for the initiation of deep convection with prevailing weak winds (e.g., Schneider et al., 2018). They are often less well resolved in operational models, which limits the forecast capabilities in contrast to situations governed by large-scale synoptic

forcing, when the forecast of precipitation is often more reliable (Baldauf et al., 2011). Previous studies have shown that the knowledge of the orographically modified flow is essential to predict intensity, location, and duration of precipitation (e.g., Rotunno and Ferretti, 2001; Rotunno and Houze, 2007).

The relevance of soil moisture for convective precipitation has been investigated in many studies (e.g., Schär et al., 1999; Findell and Eltahir, 2003; Seneviratne et al., 2010; Richard et al., 2011). Despite a robust understanding that higher soil mois-

ture leads to an increase in the near-surface specific humidity and a decrease in temperature, the soil moisture–precipitation feedback is highly complex and may vary spatially and temporally (Pan et al., 1996). Furthermore, soil moisture assumptions in models often show large differences to observations (Hauck et al., 2011). The initial soil moisture content can be of large importance as well: For a case study, Barthlott and Kalthoff (2011) showed that for drier soils (where evaporation is controlled by soil moisture), a systematic positive soil moisture-precipitation feedback exists, whereas for already quite wet soils (where

evapotranspiration is controlled by net radiation), the influence of increasing soil moisture is much weaker and the general response of precipitation to soil moisture is not systematic anymore. Additionally, the presence of horizontal land-surface wetness gradients, which induce gradients in the sensible heat flux, can foster mesoscale circulations, resulting in more precipitation over dry soils (Taylor et al., 2012). A negative soil moisture–precipitation feedback was also found for convection-resolving simulations by Hohenegger et al. (2009). In their simulations, dry initial soil moisture conditions yield more vigorous thermals



(owing to stronger daytime heating), which can more easily break through stable air barrier above, thereby leading to deep convection and ultimately to a negative soil moisture–precipitation feedback loop. Moreover, the strength of the background wind was found to change precipitation patterns even more (Froidevaux et al., 2014; Guillod et al., 2014), leading to a non-systematic soil moisture–precipitation feedback.

Besides the unclear roles of the underlying terrain and the soil moisture–precipitation feedback in different weather regimes, there are large uncertainties arising from the nonlinear character of the microphysics and the complexity of the microphysical system with many possible process pathways (Seifert et al., 2012). Many recent studies documented that the response of clouds to changes in the aerosol concentration is complex and may differ depending on the cloud type or aerosol regime or environmental conditions (e.g., Khain et al., 2008; Noppel et al., 2010; van den Heever et al., 2011; Barthlott and Hoose, 2018),

and may be complicated by processes below clouds, such as evaporation (e.g., Barthlott et al., 2017). Moreover, the validity of the invigoration hypothesis (Rosenfeld et al., 2008) in polluted conditions (i.e., updraft invigoration by additional latent heating due to a larger water load above the freezing level) and the possibility of climate responses to this effect are still considered to be open questions (Altaratz et al., 2014).

    To address predictability thoroughly, relevant sources of uncertainty need to be identified. While terrain forcing, soil mois-

ture, and aerosol impacts on convective precipitation have been investigated separately in many studies, the relative effect of these perturbations for the same weather situations has not been investigated so far. Thus, the aim of this study is to investigate the uncertainties that variations in orography, soil moisture and aerosols impose on convective precipitation by means of real-case simulations. To cover different weather regimes typical for central Europe, we analyze days with weak large-scale forcing (airmass convection) and strong large-scale forcing (passage of frontal zones). This study is unique as it is the first (to the

best of our knowledge) to address the relative impacts of these uncertainties on convective-scale predictability. It is of general relevance to assess to which extent these uncertainties should be considered in future convective-scale ensemble forecasting systems.

## 2   Methods

### 2.1   Numerical model

The general model setup follows the one from Schneider et al. (2018). All simulations were conducted with version 5.3 of the COnsortium for Small-sale MOdelling (COSMO) model (Schättler et al., 2016). It is a non-hydrostatic limited-area atmospheric prediction model, which operates on a rotated latitude/longitude grid with an Arakawa C-grid for horizontal differencing. First, simulations are performed with 2.8-km grid spacing on the operational COSMO-DE grid of the German Weather Service driven by 7 km COSMO-EU initial and boundary data (see Schneider et al. (2018) for exact domain location).

The model uses terrain-following coordinates and 50 levels in the vertical. The time integration is realized using a two-time level Runge-Kutta method (Wicker and Skamarock, 2002), the time step is set to 25 s. Whereas deep convection is fully resolved, shallow convection is parameterized with a modified Tiedtke scheme (Tiedtke, 1989). We use a 1-D turbulence scheme, which is based on a prognostic equation for the turbulent kinetic energy and can be classified as Mellor-Yamada level





2.5 (Mellor and Yamada, 1974). The model further implies a multilayer soil vegetation model TERRA-ML (Doms et al., 2011) with six soil levels. In contrast to the operationally used setup, we use the two-moment microphysics scheme of Seifert and Beheng (2006) for representing aerosol effects on the microphysics of mixed-phase clouds. The two-moment scheme predicts mass and number concentration of six different hydrometeors (cloud water, rain, cloud ice, graupel, snow and hail) and allows

to use different constant cloud condensation nuclei (CCN) concentration assumptions. The preprocessing of the initial and boundary data is done with the preprocessor INT2LM (Schättler, 2016).

Then, a 500 m grid is nested into the 2.8 km domain using one-way interfaces (Fig. 1a). Such a fine grid resolution was also used in COSMO simulations exploring the gray zone by Barthlott and Hoose (2015). They showed several benefits compared to coarser resolutions, such as a better representation of low-level convergence zones or gravity waves. The domain size is reduced

covering approximately 750×700 km (1510×1300 grid points) and spans almost entire Germany. The number of vertical levels is increased to 80, with 18 levels in the lowest kilometer. Deep as well as shallow convection are now fully resolved. Instead of a 1-D boundary-layer approximation, turbulence is now parameterized with a 3-D closure, where both vertical and horizontal turbulent coefficients are active (Doms et al., 2011). The time step is reduced to 3 s for numerical stability.

## 2.2   Sensitivity studies

To address the relative impacts of land surface and aerosol heterogeneities on deep convection, we perform several numerical sensitivity studies which are summarized in Table 1. The successive smoothing of individual terrain features is realized by taking external parameters (terrain height, land-use, roughness length etc.) at coarser resolution (1, 2.8, and 7 km), which are then interpolated onto the 500 m model grid (hereafter referred to as EXT1000, EXT2800, EXT7000). This results in somewhat lower mountain top heights and less well resolved valleys (Fig. 1). Such a technique was also applied by Schumacher et al.

(2015) for studying banded convection in the lee of the Rocky Mountains and Picard and Mass (2017) for investigating the impact of the flow direction on orographic precipitation over the US Pacific Northwest.

The majority of the sensitivity runs in this study consists of different soil moisture assumptions (Fig. 2). First, a simulation with spatially homogeneous soil moisture is performed (SM_UNI), assuming for every grid point the domain-averaged relative water content $\overline{w}_{so}$. This is done for all soil model levels to assure physical meaningful soil moisture profiles. Thus, there are

no horizontal soil moisture gradients. Then, we introduce a positive and negative soil moisture bias by increasing (SM_125) or decreasing (SM_075) the inital soil moisture field by 25 % at every grid point. The value of 25 % was selected because Hauck et al. (2011) showed that simulated and observed soil moisture in southwestern Germany differ around 20–30 %. Chess board structures are implemented with grid-lengths of 10 km (SM_10k), 56 km (SM_56k) and 112 km (SM_112k), in which moist and dry patches are regularly placed within the model domain. They represent conditions with ±25 % of the domain-averaged

soil moisture content. This technique was also applied by Baur et al. (2018) and in a similar way in large-eddy simulations by Courault et al. (2007). Similarly, random structures with a small-scale (SM_RS) or with a larger-scale (SM_RM) pattern are implemented. On average, the small-scale random pattern has a patch length similar to the 10 km chess board structure. All simulations with modified terrain and soil moisture use continental aerosol assumptions (CON) with a number density of 1700 cm$^{-3}$.



**Table 1.** Overview of the performed numerical sensitivity simulations. The relative soil moisture content $w_{so}$ is modified only at model initialization.

| Name | Description | |
|---|---|---|
| REF | original orography ($\Delta x$= 500 m) | |
| EXT1000 | smoothed orography from 1 km resolution | |
| EXT2800 | smoothed orography from 2.8 km resolution | |
| EXT7000 | smoothed orography from 7 km resolution | |
| SM_075 | reduction soil moisture by 25 % | $w_{so}$=75 %$w_{so,ref}$ |
| SM_125 | increase soil moisture by 25 % | $w_{so}$=125 %$w_{so,ref}$ |
| SM_10k | chess board structure with 10x10 km$^2$ patch size | $w_{so}$=±25 %$\overline{w}_{so}$ |
| SM_56k | chess board structure with 56x56 km$^2$ patch size | $w_{so}$=±25 %$\overline{w}_{so}$ |
| SM_112k | chess board structure with 112x112 km$^2$ patch size | $w_{so}$=±25 %$\overline{w}_{so}$ |
| SM_UNI | homogeneous soil moisture field | $w_{so}$=$\overline{w}_{so}$ |
| SM_RS | small-sized random structures | $w_{so}$=±25 %$\overline{w}_{so}$ |
| SM_RM | medium-sized random structures | $w_{so}$=±25 %$\overline{w}_{so}$ |
| MAR | maritime aerosol conditions | CCN=100 cm$^{-3}$ |
| INT | intermediate aerosol conditions | CCN=500 cm$^{-3}$ |
| CON | continental aerosol conditions | CCN=1700 cm$^{-3}$ |
| POL | polluted aerosol conditions | CCN=3200 cm$^{-3}$ |
| VAR | chess board structure with MAR, INT, CON, POL patches of 56x56 km$^2$ | CCN=1678 cm$^{-3}$ |

**Table 2.** Investigated cases.

| Day | Synoptic forcing | Characteristics |
|---|---|---|
| 30 June 2009 | weak | high pressure system over central Europe, weak mid-tropospheric winds |
| 1 July 2009 | weak | ridge over France, weak mid-tropospheric winds |
| 23 July 2013 | weak | ridge over Germany, weak mid-tropospheric winds |
| 11 September 2011 | strong | long-wave trough and low-pressure system west of the British Isles, strong mid-tropospheric winds |
| 28 July 2013 | strong | low-pressure system east of the British Isles, strong mid-tropospheric winds |
| 11 September 2013 | strong | low-pressure system over Germany, strong mid-tropospheric winds |

To address microphysical uncertainties, we introduce three other homogeneous CCN concentrations: 100 cm$^{-3}$ (maritime conditions, MAR), 500 cm$^{-3}$ (intermediate conditions, INT) and 3200 cm$^{-3}$ (polluted conditions, POL). Because aerosol concentrations are highly variable within the atmosphere (e.g., Hande et al., 2016), we also mimic a situation with spatially





varying CCN concentrations and include a chess board structure as for the soil moisture. The tiles have grid-lengths of 56 km and the concentrations of the tiles were attributed randomly, assuring that the domain-averaged CCN concentration is similar ($1678\,\mathrm{cm}^{-3}$) as in the reference simulation ($1700\,\mathrm{cm}^{-3}$, CON). The vertical CCN profile has a constant number density up to a height of 2 km and decreases exponentially above.

## 2.3 Cases analyzed

To investigate different weather regimes, we perform numerical simulations for three cases with weak synoptic forcing and for three cases with strong synoptic forcing. These are the same days already investigated by Schneider et al. (2018) who provided a detailed synoptic analysis and comparison of radar-derived and simulated precipitation totals. They showed that the model captures the overall precipitation distribution reasonably well. Thus, we only list the days and main weather characteristics in Table 2 and refer to their study for more details.

The 24 h accumulated precipitation of all reference runs (500 m original orography, unchanged initial soil moisture, continental aerosol assumption) is shown in Fig. 3. During weak synoptic forcing, the model simulates isolated convective cells with a life time of around 1–3 h (Fig. 3a-c). A more stratiform precipitation distribution is simulated for strong synoptic forcing. For these days, also embedded convection (Fig. 3d,e) and orographic precipitation enhancement (Fig. 3f, southwestern Germany) is simulated.

## 3 Results

### 3.1 Precipitation amounts and timing

The precipitation response to land surface and aerosol heterogeneities is summarized using domain-averaged precipitation totals and their deviations from the respective reference run (Fig. 4). It can be seen that the average precipitation is much smaller on weak forcing days (1.6–2.8 mm) than on strong forcing days (6.0–8.1 mm). The impact of our perturbations on precipitation deviations, however, is on average higher for weak than for strong synoptic forcing. Soil moisture and aerosols are each responsible for the maximum precipitation response for three of the cases, while the sensitivity to terrain forcing always shows the smallest spread. In general, perturbations of the orography have a larger impact during weak forcing conditions, whereas for strong synoptic forcing, the impact is rather small. This could be explained by the fact that for orographic, more stratiform precipitation, the resolution of the external data is not that important, as mesoscale rising of air on mountains can still be reasonably well simulated without detailed valley structures. Interestingly, the simulations with smoothed terrain show a systematic positive offset compared to the reference run on four out of six days. Reasons for this could be the change in near-surface temperatures, which then modify the atmospheric stability. This will be discussed in more detail in the next section.

With precipitation deviations from the respective reference run between $-12\,\%$ on 23 July 2013 and up to $+15\,\%$ on 1 July 2009, the soil moisture simulations show the highest daily variability for weak forcing cases. Furthermore, for all analyzed cases, the runs with reduced soil moisture (SM_075) always have the lowest precipitation amounts in this group of sensitiv-





ity. Positive precipitation deviations from the reference run are simulated with increased soil moisture (SM_125) indicating a positive soil moisture–precipitation feedback (except strong forcing case 28 July 2013). The impact of soil moisture on precipitation totals is generally smaller for strong than for weak synoptic forcing, which implies that land surface–atmosphere interactions are less important for weather regimes with approaching troughs or frontal systems. Different patches of dry and

wet soils have, on average, smaller effects on the simulated precipitation amounts than the dry or wet bias experiments. We therefore conclude that correct initial values are much more important than the spatial distribution of dry and wet patches assuming a constant spatial average.

The response of total precipitation to changes in the CCN concentration is more complex: In three cases (30 June 2009, 1 July 2009, 11 September 2013), the precipitation amounts decrease systematically with increasing CCN. On 11 September 2011, the

impact of different CCN concentrations is negligible. The remaining two days show a tendency towards more precipitation with higher CCN concentrations. This demonstrates the large uncertainties arising from the nonlinear character of the microphysics and the dependance of aerosol–cloud interactions on environmental conditions and cloud types. An important finding is the fact that heterogeneous CCN distribution (VAR) with a mean concentration corresponding that of the reference run (CON), can yield to precipitation deviations ranging in the same order of magnitude than changing the total CCN concentration.

Besides the integrated rain amounts, the timing of precipitation is also an important parameter for quantitative precipitation forecasting. From the precipitation rates given in Fig. 5, we see that the timing of precipitation is, at least for the domain average, not sensitive to the perturbations examined in this study. The days with weak synoptic forcing exhibit a typical summertime diurnal cycle with convection initiation around noon and largest rain rates in the afternoon. Some weaker showers exist also in the early morning hours, most probably related to model spin-up effects. In contrast, strong forcing days also show

significant precipitation amounts during nighttime. Based on the time evolution, we conclude that the different rain amounts of our sensitivity runs are mainly due to differences in rain intensity assuming that the number of simulated cells or their sizes do not differ substantially. The largest spread in precipitation rate agrees well with the largest deviations of the accumulated precipitation in Fig. 4.

## 3.2 Object-based rainfall characterization using the SAL technique

To better quantify the precipitation characteristics of our ensemble, we use the object-based structure-amplitude-location (SAL) method developed by Wernli et al. (2008). The SAL method objectively determines the characteristics of the precipitation fields by comparing the structure S, amplitude A, and location L of the simulated precipitation usually to observations for verification purposes. In this study, we apply this technique to compare the reference simulation with the rest of the ensemble, similar as in the study of Henneberg et al. (2018). The amplitude component A represents the normalized differences (between $-2$

and $+2$) of the domain-averaged precipitation values and hence gives an indication whether more (A > 0) or less (A < 0) precipitation is simulated compared to observations or a reference simulation, thereby neglecting spatial patterns. The location component L comprises two measures. First, the normalized distance between the center of mass between the objects in the reference and sensitivity simulation and second, the average distance between the center of mass of the individual objects and the total precipitation field. L can range between 0 and 1, and the smaller the value, the better the agreement. The structure



component S compares the volume of the normalized precipitation objects by capturing their size and shape. For this, the weighted means of the normalized volume of the precipitation objects are calculated. Small values indicate too small or too peaked precipitation objects compared to the reference run and large values mean the opposite. For a detailed mathematical description and examples, we refer to the paper of Wernli et al. (2008). Usually, 24 h accumulated precipitation fields are

compared with this technique with the drawback that the time evolution is not considered and errors can cancel out during the day. For this reason, we compute S, A and L values for hourly model data. These values are then averaged only for the periods with sufficient high rain intensity to avoid large SAL-errors during very weak precipitation. As the S and L components both require individual precipitation objects, we apply a threshold of $1\,\mathrm{mm\,h^{-1}}$.

  The result of this analysis is depicted in Fig. 6. The times not considered for the daily averages are marked by gray areas

in Fig. 7. The mean SAL diagram shows generally smaller SAL values as in other studies (e.g., Barrett et al., 2015). This is because we compare a reference simulation to sensitivity runs and not to observations. In particular, the location component shows small values indicating that our perturbations do not possess a large impact on the location of precipitation. The days with weak synoptic forcing generally have a larger variation in their SAL components as the days with strong synoptic forcing.

  The results of the SAL-diagrams show most variations in the structure component (Fig. 6). Averaged for all days, the aerosol

simulations have the highest absolute S value (0.15) compared to the soil moisture (0.11) or orography (0.08) simulations. The orography simulations are centered around zero S for strong synoptic forcing (Fig. 6d,e,f), which indicates that there is very little effect on the structure. This supports the previous findings, namely that changes in the terrain structure only impose a small effect on mean precipitation (Fig. 4). For the soil moisture simulations, the daily averaged S component is often negative. Whereas on weak forcing days, the individual simulations show different S values, the strong forcing cases show similar S

values for the random and chess board simulations. The aerosol simulations cover a wide range of S values, both for strong and weak synoptic forcing. Very prominent is the maritime simulation, which has the most negative S component of the aerosol simulations on all days. The reason is that the maximum precipitation amounts are much higher in the maritime than the other aerosol simulations. Since the structure scales with the maximum precipitation within each object, the S value is smaller in the maritime simulations than in the other aerosol simulations. The missing convection invigoration in our model, reflected by the

higher rain intensities and stronger updrafts in clean conditions, was also reported by Barthlott and Hoose (2018) who stated that the model results could also be influenced by the saturation adjustment scheme to treat condensational growth. Such a scheme has been shown to enhance condensation and latent heating at lower levels, which could limit the potential for a CCN increase to increase buoyancy at mid to upper levels (Lebo et al., 2012).

  The amplitude component is much smaller than the structure component, but does explain the precipitation totals well for

all strong forcing days: They show an increase in precipitation compared to the reference simulation at positive A values and a decrease for negative A values (Fig. 6d,e,f). On the weak forcing days, there are simulations, in which the amplitude does not reflect the precipitation sum. On 30 June 2009 (Fig. 6a), the EXT2800 and EXT7000 simulation have negative A components, while the precipitation is enhanced compared to the reference. On 1 July 2009 (Fig. 6b) the EXT7000 and on 23 July 2013 (Fig. 6c) the EXT2800 do not represent the precipitation totals well. Similarly, the soil moisture simulations show a good

agreement of the A component to the precipitation totals under strong synoptic forcing. On 23 July 2013, the bias simulations





show smaller absolute A values compared to the other soil moisture simulations and on 30 June 2009, the random simulations show a negative A component, while they have increased precipitation amounts compared to the reference simulation. The A component of the aerosol simulations represents the mean precipitation for weak forcing cases well, except on 30 June 2009 in the INT simulations. On strong forcing days, differences exist for example on 28 July 2013, when the A component is

positive in the POL run, but precipitation is reduced compared to the reference simulation. Considering all days, the absolute A component for the orography is 0.05 and slightly higher than that of soil moisture and orography (0.03).

The location component is generally small (Fig. 6), meaning that the place where precipitation falls is not affected much by the uncertainties addressed in our study. For the orography simulations, the shift is somewhat higher only on 30 June 2009 (Fig. 6a) and 23 July 2013 (Fig. 6c), possibly because there is a stronger surface–atmosphere coupling during weak large-scale

forcing. This would be in agreement with findings from the soil moisture simulations, as they also show higher L values for some of these day's simulations. On 28 July 2013, the bias and uniform simulation have the highest change in the location (Fig. 6e). Interestingly, the chess and random simulations show small L values, despite the formation of convergence zones due to horizontal soil moisture gradients (not shown), which could affect the location. This indicates that also other mechanisms are important to trigger convection on these days. The aerosol simulations mostly alter the location of precipitation on strong

forcing days. Interestingly, the L value is very similar for orography, soil moisture and aerosols (0.05) on all days. In summary, the amplitude and location are less affected than the structure. However, changes in the structure occur mainly due to changes in maximum precipitation amounts. Since the amplitude can explain some of the precipitation sums, we now analyze hourly time-series of the A component.

**Orography**

The daily averaged amplitude component did not explain the precipitation totals for two weak forcing days, but the time series allows for a more in-depth investigation. On 30 June 2009, the EXT7000 simulation has the highest amplitude between 12:00 UTC and 20:00 UTC, the EXT2800 simulation is slightly smaller, and the EXT1000 simulation shows the smallest values (Fig. 7a). This result fits well to the precipitation totals. After 20:00 UTC when the domain-averaged precipitation rate is below $0.02\,\text{mm}\,(30\,\text{min}^{-1})$, the amplitude becomes negative in all simulations, which can explain the daily mean amplitude.

Similarly, on 23 July 2013 the EXT1000 simulation (Fig. 7c), has positive A values between 11:00–16:00 UTC. The values become very small after 20:00 UTC, which results in a negative time-averaged A value in Fig. 4.

The fact that smoothing the orography can enhance precipitation amounts despite a weaker trigger mechanism by reduced low-level wind convergence is surprising. In the following we therefore investigate why the orography simulations show more precipitation than the reference simulation on three days (Fig. 4) by analyzing the processes underlying these sensitivities. On 28 July 2013, the deviation is small and we will restrict the analysis to 30 June 2009, as the patterns resemble those for 1 July 2009. Before 12:00 UTC the low-level wind convergence is weaker the smoother the surface is (Fig. 8b). We use the velocity

$$w_{\text{diff}} = w_{\text{max}} - w_{\text{CIN}},$$





which describes the difference between the simulated maximum vertical velocity ($w_{\mathrm{max}}$) below the level of free convection and the required updraft to overcome convective inhibition ($w_{\mathrm{CIN}} = \sqrt{2 \times \mathrm{CIN}}$) to investigate whether convection can be initiated or not. If $w_{\mathrm{diff}}$ is positive, the updrafts are strong enough to transport air parcels to their respective level of free convection, convection will be initiated and CAPE released (Trier, 2003). The combined measure of gridpoints with $w_{\mathrm{diff}} > 0\,\mathrm{m\,s^{-1}}$

and CAPE $> 600\,\mathrm{J\,kg^{-1}}$ (Fig. 8c) confirms our expectations, namely that it is more difficult to initiate deep convection with smoother surface due to reduced low-level wind convergence. As a consequence, there is a short delay in precipitation initiation and hence CAPE has more time to build up through solar heating (Fig. 8a), especially in the EXT2800 and EXT7000 simulations. Despite less favorable conditions, low-level wind convergence is still strong enough to trigger convection in these simulations. Because the static instability is higher, convection is stronger with more precipitation than in the reference simu-

lation. The precipitation difference between the reference and the EXT1000 simulation is only minor, and so are differences in CAPE, possibly because the difference in terrain height is also marginal.

**Soil moisture**

The A component is important to quantify the precipitation totals. However, on 30 June 2009 the daily-averaged A component does not follow the precipitation totals in the simulations with random patterns. As can be seen in Fig. 7g, their values become

rather small after 18:00 UTC, which mainly determines the daily average in Fig. 6. On 1 July 2009, the precipitation was reduced in the SM_075 simulation compared to the reference case, but the daily-averaged amplitude was close to zero. Similarly, on 23 July 2013, the bias simulations showed a strong positive soil moisture–precipitation feedback but the daily-averaged sign in A was similar as for the random simulations. We will investigate the patterns for 23 July 2013 as they are most pronounced. Interestingly, the time series of the A component shows changes in sign for all simulations (Fig. 7i). While the wet

run (SM_125) shows negative values around noon, all other runs reveal positive values. Around 15:00 UTC, there is a change in sign for all model runs. Thus, the daily averaged A value is not representative. The temporal evolution of the A component fits relatively well to the temporal evolution of precipitation and can be explained by convection-related parameters (Fig. 9). The soil moisture controls the partitioning of the available energy at the surface (net radiation minus soil heat flux) into latent and sensible heat. During daytime, the Bowen ratio $\beta$ increases to values larger than 1 in the SM_075 simulation as a result

of the dominating sensible heat flux. This enhances the near-surface temperature (not shown) and turbulence in the boundary layer, which will lead to an increased low-level wind convergence compared to a simulation with enhanced soil moisture. Therefore, the lifting condensation level is higher (not shown) and CAPE is reduced compared to the SM_125 simulation. This is represented by a positive A (Fig. 7i). The enhanced low-level wind convergence in the SM_075 simulation is connected with stronger updrafts, thereby triggers convection leading to more precipitation between 10:00–14:00 UTC compared to the

reference or SM_125 simulation. On the other hand, CAPE can build up higher in the SM_125 simulation, and this leads to an enhancement of precipitation compared to the reference simulation after 15:00 UTC. The higher precipitation rate compared to the reference simulation is reflected in the increase of the A value (Fig. 7i) and leads even to a positive A component in the SM_125 simulation after 18:00 UTC. The random and chess board simulations show increasing values until 12:00 UTC, remain positive until 15:00 UTC and decrease afterwards to negative values. These mass process rates have been integrated



vertically and averaged over the domain. In general, these simulations show similar values for the Bowen ratio and CAPE, and only minor differences in the low-level wind convergence compared to the reference run. This leads to small differences in precipitation, which results in differences in the amplitude.

The simulations for days with strong synoptic forcing show less variations in the A component than do the days with weak forcing. On 11 September 2011 and 2013, the A component shows only small differences in all model runs. Only on 28 July 2013, larger deviations from 0 exist for the soil moisture and aerosol uncertainties (Fig. 7k,q). The precipitation totals are in agreement with the evolution of the amplitude component on all days. As has been noted earlier, this day is the only one without a systematic soil moisture–precipitation relationship. Before 15:30 UTC, both the dry and the wet simulations mostly reveal higher amplitude components as the reference run. Later on, both time series become negative, resulting in an overall precipitation reduction compared to the reference run. In general, the surface fluxes are smaller on strong forcing days and hence the surface–atmosphere coupling is weaker. Changes that do occur mainly result from modifications in the total precipitable water as a result of small changes of evaporation (not shown).

**Aerosols**

On 30 June 2009 (Fig. 7m) between 9:00–19:00 UTC, on 1 July 2009 (Fig. 7n) between 9:00–14:00 UTC and 11 September 2013 (Fig. 7r), there is a tendency for decreasing amplitude from maritime to continental conditions, and so does the precipitation amount (Fig. 4). One common characteristic for these days is that the domain-averaged updraft velocities within the clouds (regions were the integrated cloud and rain water path is larger than $0.01\,\mathrm{mg\,m^{-2}}$), is always smaller than $0.25\,\mathrm{m\,s^{-1}}$ (not shown).

On 11 September 2011 there is a weak decrease in the amplitude component after 12:00 UTC from polluted to maritime conditions (Fig. 7p). On 28 July 2013, the amplitude is highest in the polluted and lowest in the maritime simulation (Fig. 7j). On 23 July 2013, the order changes at 15:00 UTC (Fig. 7o) and also the precipitation sums are less systematic. On these three days, the updraft velocities within clouds are always higher than $0.38\,\mathrm{m\,s^{-1}}$ and therefore higher as in the three cases described above. The different vertical velocities, and thus the environmental conditions then affect the dominant cloud microphysical pathways, which are now analyzed using microphysical process rates. The warm-phase processes are autoconversion (collision of cloud droplets) and accretion (rain droplets collecting cloud droplets), the dominant cold-phase processes are vapor deposition on ice crystals and riming (collision of a droplet and an ice crystal). In general, cold-rain processes dominate in all our simulations as the ratio of warm- to cold-rain processes is always less than one (Fig. 10). On 28 July 2013 and 11 September 2011, cold processes are much more important than warm processes as indicated by the small ratio of warm- to cold-rain processes, due to the stronger lifting. On 23 July 2013 the ratio is higher, possibly because we find a regime change during the high intensity period. For these three days, the higher vertical velocity leads to a pronounced transport of cloud droplets towards higher altitudes, especially for polluted conditions, when cloud droplets are smaller (not shown) and hence persist longer within the clouds than it is the case for maritime conditions. As mentioned earlier, we do not observe stronger updrafts in polluted conditions, and thus no convection invigoration as hypothesized by Rosenfeld et al. (2008). Instead, when the cloud particles grow via the cold phase and then precipitate, they have bigger sizes than the droplets in the maritime con-



ditions (not shown), and are thus less susceptible to evaporation below cloud base which leads to higher precipitation amounts
with increasing CCN.

On the other hand, warm phase processes are almost similarly important as cold phase processes on 30 June and 1 July
2009, due to the weaker updrafts and even on 11 September 2013, the ratio is always above 0.5. On these days, the suppression
of collision-coalescence with increasing CCN has larger effects on the precipitation amounts (as cold-phase processes and
melting contribute relatively less) and hence a reduction in precipitation towards more polluted conditions. For a more detailed
analysis of hydrometeor profiles and microphysical process rates, we refer to Schneider (2018).

## 4 Summary and conclusions

The purpose of this study was to investigate the relative contribution of orography, soil moisture and aerosols on the pre-
dictability of deep convection. To this end, we performed $500\,\mathrm{m}$ grid length numerical simulations with the COSMO model
for six real-case events over Germany classified into weak and strong large-scale forcing. The sensitivities comprise smoothing
the terrain, systematic changes in the initial soil moisture field, and different homogeneous and spatially heterogeneous CCN
concentrations.

In general, weak forcing days show smaller precipitation amounts than strong forcing days, but a higher precipitation sus-
ceptibility ($-12$ to $+15\,\%$) to the applied changes than strong forcing days ($-9$ to $+7\,\%$). We find that uncertainties in soil
moisture and CCN concentrations contribute the most to the ensemble spread. The modifications in soil moisture have the
strongest impact on two weak forcing and one strong forcing day. For the majority of the analyzed cases, the model produces
a positive soil moisture–precipitation feedback in agreement with e.g., Findell and Eltahir (2003) or Cioni and Hohenegger
(2017). Different patches of dry and wet soils have, on average, smaller effects on the simulated precipitation amounts than
the dry or wet bias experiments. We therefore conclude that correct initial values are much more important than the spatial
distribution of dry and wet patches assuming a constant spatial average. The aerosol simulations have the strongest impact on
one weak forcing and two strong forcing days. Furthermore, we find that an increase in CCN concentrations can either lead to
an increase or decrease in precipitation, depending on the environmental conditions and different contributions of warm and
cold-rain processes. In all our simulations, the contribution of cold-rain processes is higher than that of warm-rain processes.
For weak updrafts, however, the relative role of the warm-phase processes is higher and a reduction in precipitation occurs
with higher CCN concentrations and smaller droplets. For stronger updrafts, the cold-phase processes dominate. The precipi-
tation thus increases with increasing CCN, as bigger raindrops that occur via the cold-phase are less susceptible to low-level
evaporation (Tao et al., 2007; Barthlott et al., 2017). An important finding is the fact that a heterogeneous CCN distribution
with a mean concentration corresponding to that of the reference run (continental assumption), can yield to precipitation de-
viations ranging in the same order of magnitude than changing the total CCN concentration. The fact that soil moisture and
aerosol perturbations contribute in a similar magnitude to the precipitation totals suggests that aerosols are indeed important for
quantitative precipitation forecast (Miltenberger et al., 2018). The smallest deviations from the reference runs occurred when
introducing orography uncertainties. Surprisingly, on three days, the smoothing of terrain features lead to higher precipitation




amounts. This could be attributed to a slightly increased instability compensating for the weaker triggering by low-level wind convergence. In addition, the resolution of external data is less important for strong synoptic forcing as mesoscale rising of air over mountain ridges can still be reasonably well simulated without fine-scale orographical features like valleys.

To investigate amplitude, location and structure of precipitation, we compute SAL diagrams based on hourly precipitation fields. We find that the structure parameter is affected the most, followed by the amplitude and only small variations in the location. On average, the highest structure parameters occur in aerosol simulations (absolute mean 0.15). Changes in the structure occur mainly due to increased maximum precipitation amounts. The evolution of rain intensities was mostly well correlated with the amplitude component. The location component does not vary much between the three sensitivities and the absolute value lies around 0.05. Because of this resemblance, we hypothesize that this shift is due to noise resulting from different CCN assumptions and initially small perturbations to the thermodynamics/dynamics. This is in accordance with previous findings of Schneider et al. (2018), namely that the shift in precipitation in the orography simulations resembles the patterns for artificially introduced noise. As a thorough discussion of all involved processes and feedbacks for all sensitivities and cases would be exhaustive, we refer to Schneider (2018) for more details.

To increase the reliability of operational ensembles, we will probably observe a further increase in the use of ensemble methods, but this will require more effort to perturb the model (Leutbecher et al., 2017). The overall goal for these perturbations is to make them as realistic and relevant as possible. For the operational forecast, ensembles, which perturb initial conditions, boundary data and model physics, are run to account for the various uncertainties. Based on the results of this study, we suggest to account for variations in soil moisture and aerosols, also because both are associated with a high measurement uncertainty (e.g., Van Reken et al., 2003; Hauck et al., 2011). For the soil moisture perturbations, adapted ensembles could be meaningful, i.e., by perturbing different components depending on the large-scale synoptic situation. After all, we conclude that these uncertainties should be included in a full ensemble forecasting system containing other key sources of uncertainty to estimate their relative importance for longer periods.

*Data availability.* COSMO model output is available on request from the authors.

*Author contributions.* CB and CH developed the project idea, all authors designed the numerical experiments and LS carried them out. LS conducted the analyses and all contributed to the interpretation of the results. LS wrote the first version of the paper, CB edited it with contributions from CH.

*Competing interests.* The authors declare that they have no conflict of interest.




*Acknowledgements.* The research leading to these results has been done within the subproject "B3: Relative impact of surface and aerosol heterogeneities on the initiation of deep convection" of the Transregional Collaborative Research Center SFB/TRR 165 "Waves to Weather" funded by the German Research Foundation (DFG). The authors wish to thank the Deutscher Wetterdienst (DWD) for providing the COSMO model code and the initial and boundary data. This work was performed on the computational resource ForHLR I and II funded by the
5   Ministry of Science, Research and the Arts Baden-Württemberg and DFG ("Deutsche Forschungsgemeinschaft").





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





**Figure 1.** Reference orography at 500 m grid spacing (a) and interpolated orography from 1 km (b), 2.8 km (c) and 7 km (d) to the model grid. The black rectangle depicts the 500 m simulation domain.

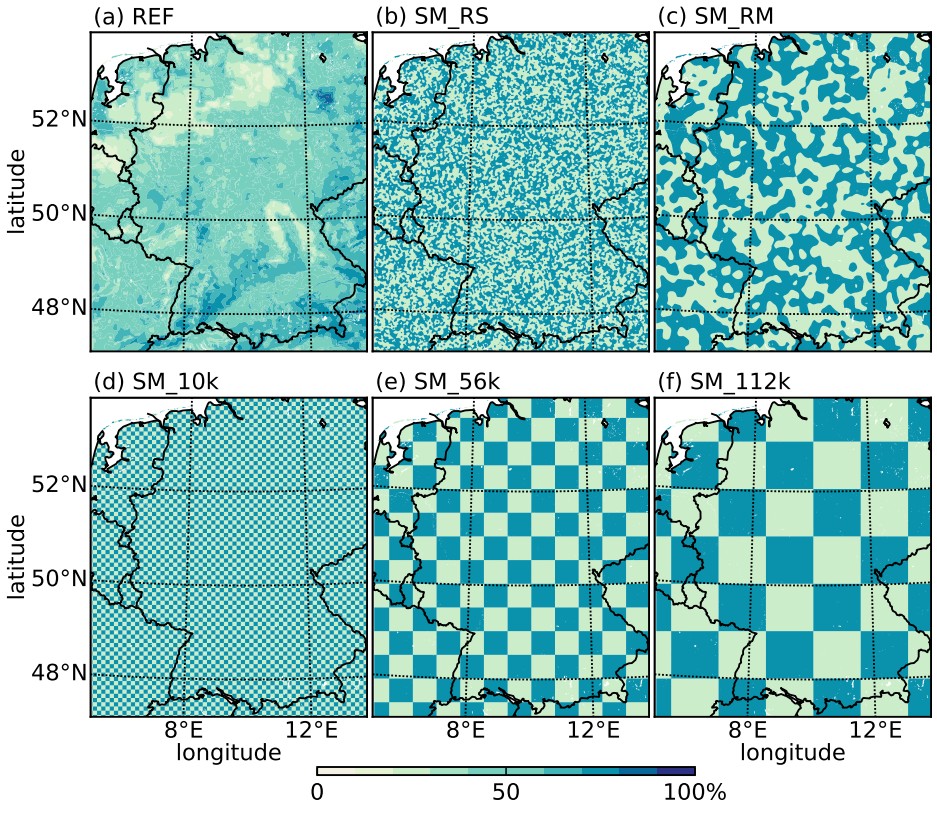

**Figure 2.** Relative water content of the reference field (REF, a), the simulations with small (SM_RS, b) and medium (SM_RM, c) random structures, and chess board structures with grid length of 10 km (SM_10k, d), 56 km (SM_56k, e) and 112 km (SM_112k, f).





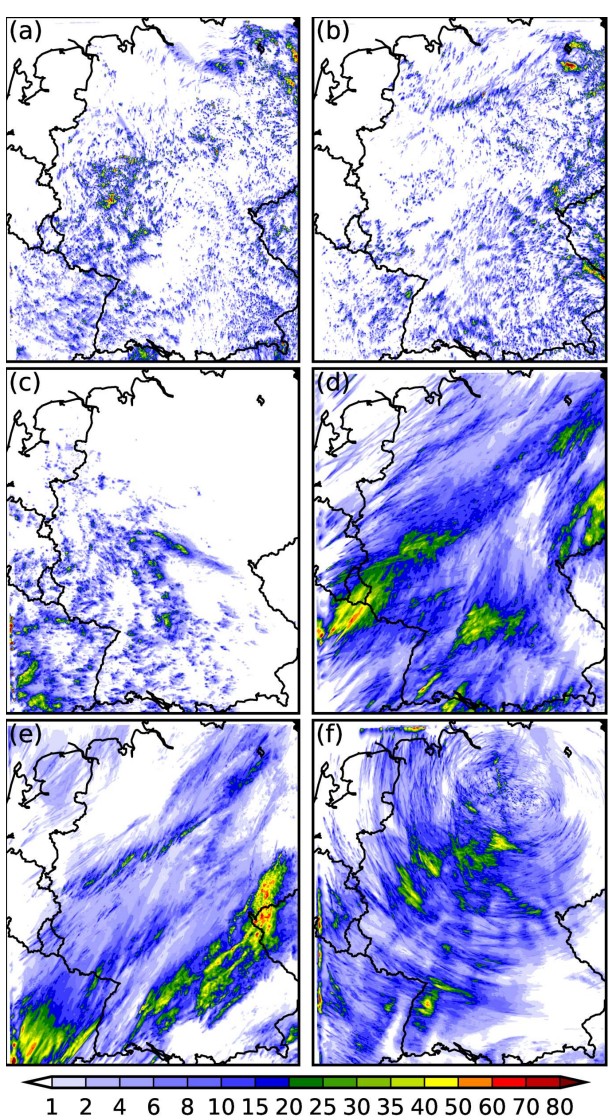

**Figure 3.** 24 h precipitation amount of 500 m grid length reference run in mm for the six days of investigation: (a) 30 June 2009; (b) 1 July 2009; (c) 23 July 2013; (d) 11 September 2011; (e) 28 July 2013; (f) 11 September 2013. Figure adapted from Fig. 5 in Schneider et al. (2018).





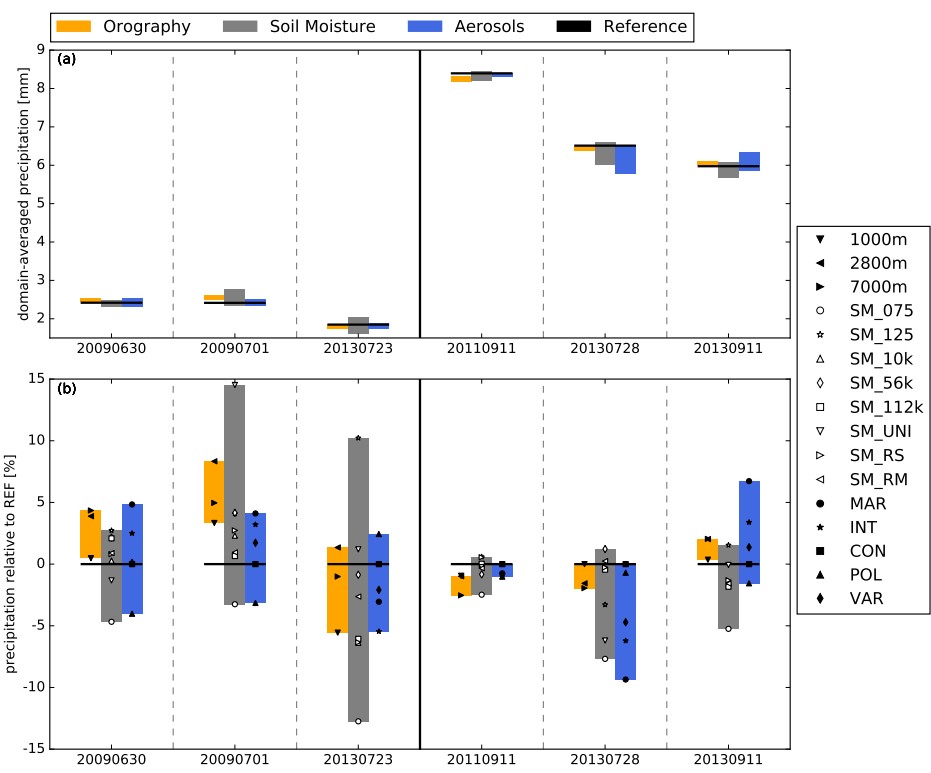

**Figure 4.** 24 h domain-averaged precipitation (a) and deviation from the respective reference run (b) for the six days of investigation. The symbols denote the precipitation deviation and the height of the bar shows the distance between minimum and maximum mean precipitation for each set of sensitivities.





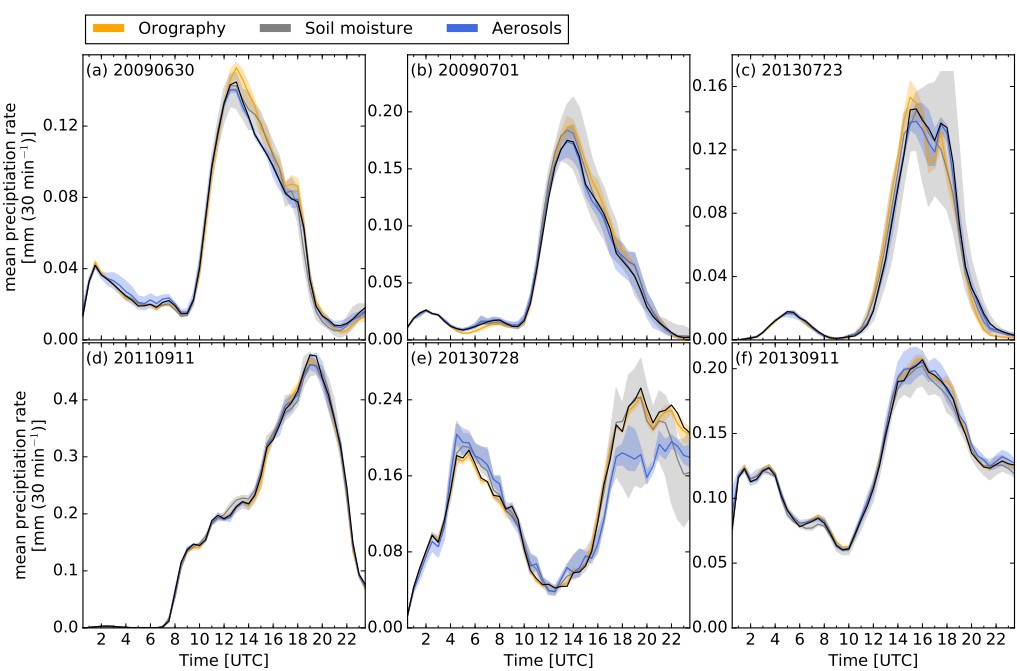

**Figure 5.** Domain-averaged precipitation rates for the six days of investigation. The color-coded areas represent the spread, i.e. the maximum and minimum mean precipitation for each set of sensitivities.





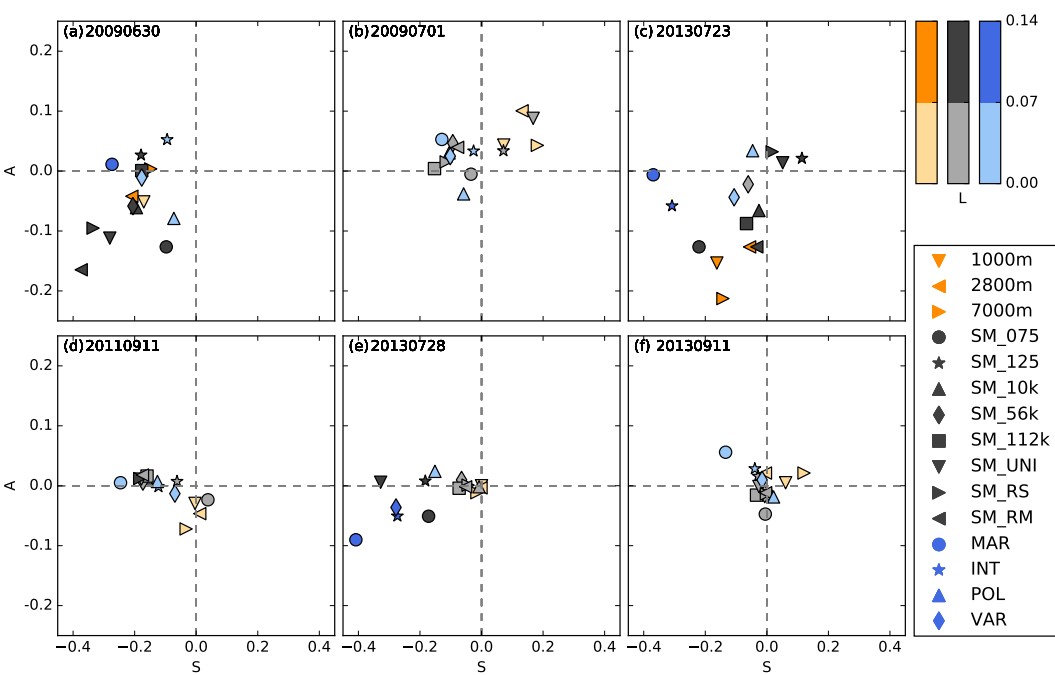

**Figure 6.** Mean SAL diagram showing the structure, amplitude, and location component for the six days of investigation averaged from hourly rainfall amounts. The more the values deviate from 0, the larger is the deviation from the respective reference run. An identical prediction would have values of 0 for each component. Note the different axes for A and S.





**Figure 7.** Time series of hourly computed A values for the simulations with modified orography (a)–(f), soil moisture (g)–(l) and aerosols (m)–(r) for the six days of investigation. Gray shaded areas are excluded from the computation of daily mean values in Fig. 6.





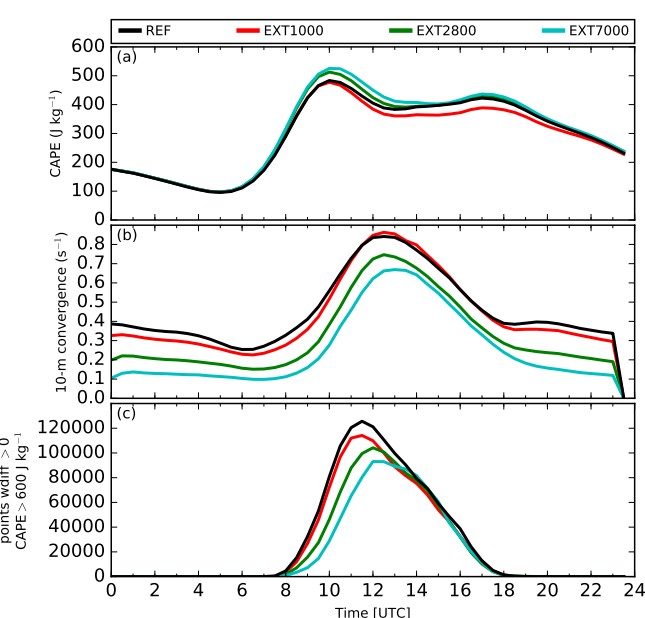

**Figure 8.** Temporal evolution of CAPE, 10 m wind convergence and number of points with $w_{\mathrm{diff}} > 0 \, \mathrm{m \, s^{-1}}$ and CAPE $> 600 \, \mathrm{J \, kg^{-1}}$ on 30 June 2009 in the orography sensitivity experiments.





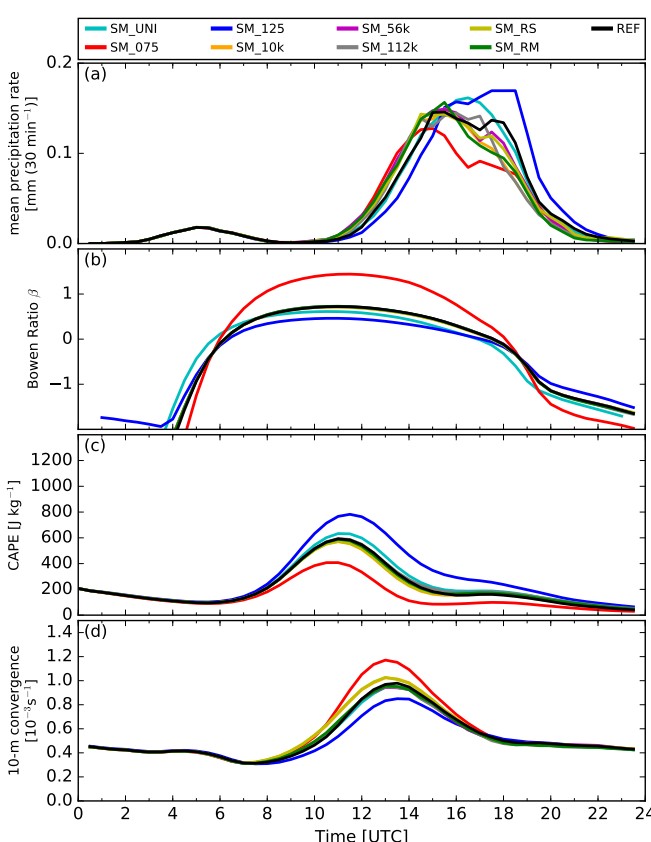

**Figure 9.** Temporal evolution of mean precipitation rate, Bowen ratio, CAPE and 10-m wind convergence on 23 July 2013 in the soil moisture sensitivity experiments.



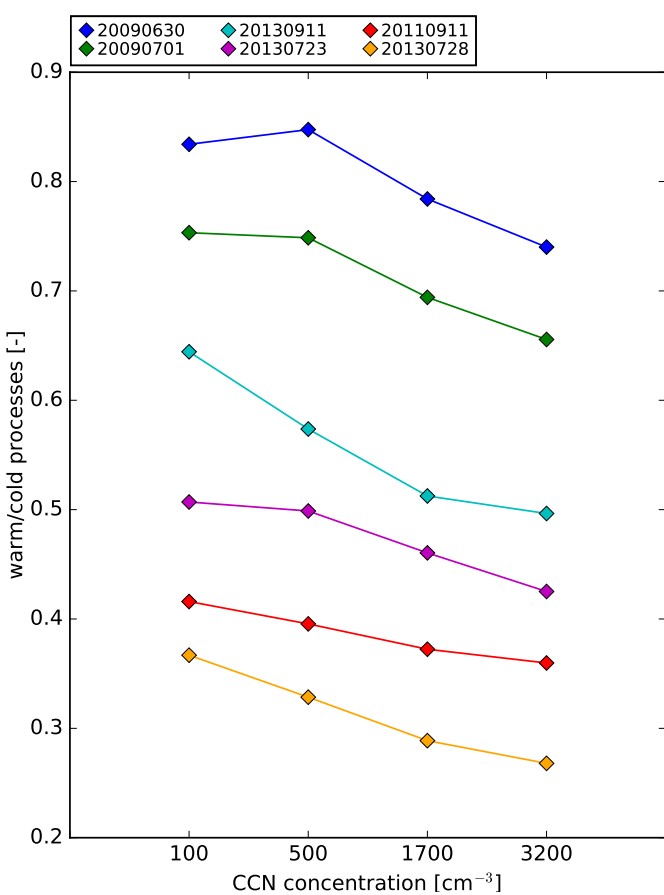

**Figure 10.** Ratio between warm (autoconversion and accretion) and cold (vapor deposition and riming) rain processes as a function of the CCN concentration.