# Peer review of "Relative impact of aerosol, soil moisture and orography perturbations on deep convection"

_Atmospheric Chemistry and Physics, 2019_

## Referee Comment (RC1) · Anonymous Referee #1 · 3 May 2019

The study investigates the difference between short-term cloud-resolving weather forecast type simulations that have been perturbed in their specification of aerosols, soil moisture and the underlying orography. This is done for six different cases. While three of the cases cover weak synoptic forcing, the other three address strong synoptic forcing. The difference between the simulations is quantified in terms of the 24h domain-averaged precipitation sum, the evolution in time of the domain mean precipitation rate and the Structure Amplitude and Location (SAL) parameter. The authors find that differences between the simulations are larger for weak synoptic forcing, than for strong synoptic forcing. Moreover, the soil-moisture perturbations lead to the biggest differences between the simulations, with a positive soil-moisture precipitation feedback. Perturbations to the prescribed aerosol concentrations can have an equally large

impact on the simulated precipitation field, but a reduction or increase of aerosols can either lead to an increase or decrease of precipitation, and vice versa. Changes to the orography showed the smallest impact on the resulting precipitation.

The study addresses interesting aspects and is well written. Yet, in parts it lacks precision about the exact aims and methods to address the open questions. Thus, in my view it needs major revisions before it can be published.

General comments:

1. The authors claim to investigate the contribution of soil moisture, aerosols and orography on the predictability of deep convection. Yet, they investigate the impact of these factors on 24-hour sums of precipitation. In order to address predictability, further analysis would be needed, as e.g. the RMSE between the simulations, the growth of the RMSE over time, and the saturation of the error. I suggest removing the term "predictability" from the manuscript. Moreover, the term "ensemble spread" is used. Yet, the different experiments are introduced as "sensitivity studies", but not as "ensemble perturbations".

2. Page 4, sensitivity studies: The perturbations of soil moisture are applied to the relative water content w_so, irrespective of the soil type. Dependent on the soil type this results in a different perturbation. It would be better to first compute the soil-moisture index/soil moisture range/fraction of available water (theta-theta_wp)/(theta_fc-theta_wp), to then perturb this quantity, and to convert back to relative water content, as this would yield a fairer perturbation across soil types. Is the field capacity reached in the simulations with increased soil water content?

3. The smoothing of the orography not only modifies the static stability of the atmosphere, but also the moisture content by including/excluding parts of the moisture-laden lower troposphere. CIN/CAPE is one measure to predict the amount of convection that will occur. Another measure is the amount of moisture stored in the atmosphere that can be tapped by the convection. I suggest to include a plot showing the integrated precipitable water before the onset of convection for the different sensitivity experiments. On page 11, lines 11-12 it is stated that the total precipitable water changes. I think it is important to show these plots and to interpret the results. Page 1, line 20: "but that precipitation amount depends ..." in my opinion it is not only the strength of the trigger mechanism, but also the amount of moisture available for precipitation formation.

4. Page 4, sensitivity studies: Please describe the generation of the random patterns in more detail.

5. Page 5, lines 5-7: "we therefore conclude that ..." in my opinion this is not a valid statement as apples are compared with pears. The perturbation that is applied in the simulations with the altered soil-water content is much larger than just shifting around the soil water, but not changing its magnitude.

6. Page 8, lines 23-24: "since the structure scales ..." what does this imply for the S values in the other sensitivity tests?

7. Paragraph 3.2: within this paragraph an attempt to validate the SAL metric is mixed with results on the different perturbation techniques. This makes it hard to read the paper. The manuscript would benefit if you split the text into an evaluation part and a results part.

8. Page 9-10, section Orography: In the text the development of the grid-scale (resolved) convection is discussed. Yet, a parameterization scheme for shallow convection is active. When is sub-grid scale convection initiated in the different experiments, and how much of the instability is being eroded by the shallow convection parameterization scheme?

9. How long are the simulations run for? Are there any spinup effects? Are the differences largest during the initial phase, or does the difference saturate, or even decrease, once the model has re-adjusted after the initial perturbation?

Specific comments:

- Page 2, line 11: I suggest to replace "uncertainties" by "states" or "conditions", as no analysis of the underlying uncertainties is given.

- Page 3, line 1: replace "barrier" by "barriers", or include "a" before "stable"

- Page 3, lines 14-22: to my knowledge there are two papers addressing the combined effect of soil-moisture perturbations and orography: Rihani et al. (2015), and Imamovic et al (2017).

- Page 4, line 1: replace "implies" by "includes"

- Page 4, line 13: replace "coefficients are active" by "diffusion is active"

- Page 4, line 24: replace "physical" by "physically"

- Page 7, line 12: replace "dependancy" by "dependency"

- Page 7, line 13: include "a" before "heterogeneous"

- SAL technique: I suggest to use bold, italic or calligraphic letters for A, L, or S in the text.

- Page 8, line 10: replace "as" by "than"

- Page 11, line 9: replace "as" by "than"

- Page 11, Aerosols section: what about the collection processes of the falling cold hydrometeors, e.g. self-collection of graupel and hail, collection of rain by graupel or hail?

- Page 12, line 19-21: "we therefore conclude that...more important...", more important for what? As stated earlier, I don't think that this is a very meaningful statement.

- Page 12, line 29: either remove "to" before "precipitation", or replace "yield" by "lead"

- Page 12, line 32: replace "forecast" by "forecasting"

- Figure 4: the symbols indicating the different sensitivity runs are very hard to read.

- Figure 6: I suggest to include the reference simulation at (0,0)

Rihani, J. F., F. K. Chow, and R. M. Maxwell. 2015. Isolating effects of terrain and soil moisture heterogeneity on the atmospheric boundary layer: Idealized simulations to diagnose land-atmosphere feedbacks. Journal of Advances in Modeling Earth Systems 7, 915-937.

Imamovic, A., L. Schlemmer, and C. Schär. 2017. Collective impacts of orography and soil moisture on the soil moisture-precipitation feedback. Geophysical Research Letters, 44, doi:10.1002/2017GL075657.
* * *

---

## Referee Comment (RC2) · Anonymous Referee #2 · 11 May 2019

The manuscript "Relative impact of aerosol, soil moisture and orography perturbations on deep convection" investigates different aspects that affect precipitation stemming from deep convection in numerical weather forecast simulations for six cases with different synoptic forcing. The study includes the impact of cloud condensation nuclei (CCNs) concentration and distribution, soil moisture content and distribution and the smoothing of orography. The analysis comprises domain averaged precipitation sums, time development of precipitation averages, time averaged values of the structure, amplitude and location parameters and time resolved values of the amplitude parameter. Furthermore, convective parameters and cloud conversion rates are analysed. This analysis showed that the introduced soil moisture and CCN concentration modifications affect precipitation amounts stronger than the applied changes in orography. While soil

moisture causes a positive feedback on precipitation, CCN concentrations affect precipitation unsystematically. The study is very comprehensive and covers many aspects which can affect precipitation rates in numerical model simulations. Despite the effort which was put in the model simulations, the motivation for the conducted simulations is not clear and the analysis methods need to be improved. Thus I recommend the study for publication after major revisions.

General comments: The results from simulation with varying soil moisture, orography (including other external data) and CCN concentrations were compared to each other and conclusions that soil moisture and CCN concentration can affect precipitation significantly were drawn. However, it is not clear to me, how the variation in the initial model conditions compare to each other. Soil moisture differences compare to differences which can occur between observed and modeled soil moisture content. CCN concentrations are described as varying between maritime and polluted conditions. How does this compare to observations? Modification in the orography are based on smoothing to coarser model resolution. How strong do these changes vary eq. compared to variation which can be achieved by tuning the orographic smoothing? Instead of addressing the question on how the resulting precipitation differs, I would rather ask the question how much one variable (eq. aerosol) needs to be changed to achieve the same model spread in precipitation as by a change in soil moisture content by eq. 25%. In the end the fact that soil moisture affects precipitation strongest can also be a result of the strength by which the soil moisture was modified. Preferable to do this an independent ensemble spread would provide a first estimate or the comparison to observational data which are present from the previous study from Schneider. Also possible to use the coarse scale simulation as reference comparison. While a lot of effort was put in conducting a very comprehensive model study, the analysis is lacking and often confusing. For example, mean values of the SAL score are calculated which are later revised as the A value is not representative. Page 10 Line 13 to 22 describe why the previously used values of A are not representative

**ACPD**
Specific comments:

Abstract: Page 1 Line 13: correct initial values are much more important than the spatial distribution ... What is meant by correct initial values? I think what is meant is something like: The amount of soil moisture affects precipitation stronger than its spatial distribution.

Page 1 Line 18: ... that the structure component is highest in the soil moisture ... Are structure values really higher for all soil moisture and CCN simulations or is structure most sensitive to changes in soil moisture (content) and CCN concentration. Also, I would avoid talking about structure in the abstract as the concept of SAL is not clear to the reader yet. What does structure mean? – Intensity of precipitation?

Page 1 Line 19: ... dominant mechanism for convection initiation ... trigger mechanism What are these mechanisms?

Page 1 Line 20: Location and amplitude parameters are both much smaller. Change to: Location and amplitude parameters both vary over a much smaller range.

Introduction: Missing explanation about the soil moisture precipitation feedback. Missing: The conclusion of the study is to include soil moisture and aerosol variation in ensemble forecast to achieve a sufficient model spread? An overview of how ensemble spreads are generated in current ensemble setups would be beneficial.

Page 2 Line 26: What is meant by soil moisture assumptions? Assumptions in the formulation in the couple soil model or assumptions about the soil moisture content?

Page 3 Line 17: Aerosol or CCN? I think the study is constrained to changes in the CCN concentration while the freezing /nucleation scheme remains untouched?

Methods:

Why chess board like structure to modify soil moisture field. That looks like a very artificial change and could cause artificial circulation. Is the domain average soil moisture
content the same? How are the random perturbations generated?

How exactly are the inhomogeneous CCN concentration generated? Even though, CCN concentration can vary spatially I would assume they are advected similar to the clouds (at least in strong forcing cases). Clouds which travel through region with strongly varying aerosol conditions seem to be rather unrealistic to me and I fear this causes unphysical affects? In weak forcing cases I would assume the effect of this CCN perturbation is randomly depending on where convection is triggered with respect to the CCN modification.

I assume the changes in the smoothing of external data affecting the orography is only relevant in regions with complex topography, while the modifications in soil moisture and CCN concentrations are applied across the whole model domain.

Page 4 Line 24: Change to homogenous soil moisture is done for all model levels. From that it is not clear if all soil moisture perturbations are applied on all soil layers.

Table 1: Maybe I missed it: what is the aerosol concentration in the reference run? In the text it says simulations with modified terrain and soil moisture run with 1700 cm-3. Is this also the case for the reference run? If so, I wonder what the difference to CON is.

The simulations were compared to radar observations in a previous study. The present study would highly benefit from including this comparison eg. also by using SAL analysis. This provides a reference deviation for a better quantification of the variations in precipitation that occur from different model settings.

Results: For the analysis SAL was used. While the A value mainly describes the changes in precipitation amount, it does not give much more information as the comparison of the precipitation amount. However, the S value gives information of whether precipitation becomes more intense and locally constrained (deeper convection) or increases in size. For that it is important to have A as comparison in order to derive if a
change in structure is caused by a change in the area or intensity. This connection was not drawn in the analysis. Further, the averaging over SAL values over time diminish some effects. I'm not sure what the worth of an averaged SAL values are. Especially later the relevance and correctness of the A-value is often questioned.

Page 6 Line 17: ... the sensitivity to terrain forcing always shows the smallest spread. As already mentioned above, I find this hard to judge as the sensitivity highly depends on the strength on variation.

Page 6 Line 25: How does stratiform precipitation match the title?

Page 8 Line 2-3: Change small to negative Change too small to smaller. Too smaller sound as if this is wrong but it is just different to the reference case. Change large to positive.

Page 8 Line 11: Are the SAL values smaller compared to other studies because of the model to model comparison or because of the averaging?

Page 8 Line 29: On weak forcing days, there are simulations, in which the amplitude does not reflect the precipitation sum? So, what is the sense of the previous analysis than. This makes it really difficult to follow. Also on Page 10 Line 21.

Page 10 Line 7: Change especially to only.

Page 10 Line 24: Bowen ration not introduced yet. What does Bowen ration above 1 mean? Higher latent or sensible heat?

Page 10 Line 28: What is the relation between CAPE and A-component? Why does precipitation increase with reduced CAPE? (If my interpretation of positive A is correct). In Line 30 the argument is, that enhanced CAPE enhances precipitation. In this argumentation I miss arguments about the changed moistening of the atmosphere, what is needed to trigger, convection, destabilize the atmosphere and also to provide enough moisture for precipitation. Showing some more results about convective parameters such as surface temperature, CAPE or LCL developments may support the

**ACPD**
argumentation.

Page 12 Line 1: What kind of precipitation is found below cloud base (rain or snow). Is it only a size argument what makes them less susceptible to evaporation (or sublimation)?

Page 12 Line 20: What are correct initial values?

Technical comments: Page 2 Line 10: the state of the atmospheric Atmosphere or atmospheric condition

Page 11 Line 27: switch 28 July and 11 September as in the text above 28 July is also mentioned first

---

## Author Comment (AC1) · 7 Jun 2019

**Responses to the reviewers**

Relative impact of aerosol, soil moisture and orography perturbations on deep convection

by Linda Schneider et al. June 7, 2019

We thank both reviewers for reading the manuscript and providing detailed comments. We have carefully considered all comments and changed the manuscript accordingly. Please find below our responses in blue.

We also added Andrew I. Barrett to the list of authors, as he contributed with technical implementations to write out microphysical process rates and to use horizontal heterogeneous CCN concentrations in the COSMO model code. In addition, he was involved in the interpretation of the results.

**Reviewer 1**

General comments:

1. The authors claim to investigate the contribution of soil moisture, aerosols and orography on the predictability of deep convection. Yet, they investigate the impact of these factors on 24-hour sums of precipitation. In order to address predictability, further analysis would be needed, as e.g. the RMSE between the simulations, the growth of the RMSE over time, and the saturation of the error. I suggest removing the term "predictability" from the manuscript. Moreover, the term "ensemble spread" is used. Yet, the different experiments are introduced as "sensitivity studies", but not as "ensemble perturbations".

   We agree with the reviewer that an analysis of the RMSE would provide a valuable information to the reader. We therefore made those calculations and inserted a new figure at the end of section 3.1 together with new text.

   **Changes to paper:**

   *"To further address deviations of the sensitivity runs from the reference run, we now analyze the root mean square error (RMSE) of total precipitation and its temporal evolution (Fig. 6). It can be seen that the increase of errors is generally largest for the times with maximum precipitation rates. For the weak forcing cases (Fig. R.1 (a)-(c)), orography and aerosol modifications lead to larger RMSE values already for smaller rain rates in the early morning hours than the soil moisture runs. Interestingly, the soil moisture runs show a steeper increase once convection is initiated around 10:00–11:00 UTC. In agreement with recent findings of Baur et al. (2018), this indicates that heterogeneous soil moisture perturbations mainly influence the convection initiation via secondary dynamical effects (like thermally induced circulations), whereas CCN and orography variations induce variability already from the beginning of the simulation. Overall, the errors are largest in the soil moisture and orography runs and smallest in the aerosol runs. This is also true for the cases with strong synoptic forcing (Fig. 6 (d)-(f)). However, there is no distinct temporal delay of the soil moisture runs indicating that its influence on precipitation initiation is weaker than on days with weak synoptic forcing. We also find that the spread at the end of the simulation of the aerosol runs is always higher for strong than for weak synoptic forcing, which points to a larger role of CCN concentrations in this weather regime. The same holds true for the soil moisture runs which also possess the largest spreads of all sensitivities studied here. On average, the orography runs have a similar spread in both weather regimes."*

[Figure]

Figure R.1: Root mean square error (RMSE) for precipitation of the sensitivity runs compared to the reference run. Thick lines indicate the mean and color-coded areas the range between the minimum and maximum RMSE.

Concerning the terms predictability, ensemble, sensitivity:
This study deals with the predictability of deep convection and explores new techniques to disturb the model. We are aware that the ensemble size is rather small and that the terms predictability and sensitivity have to be chosen with care. We therefore decided to keep the term predictability in the manuscript (5 occurrences in Abstract, Introduction, Summary).

**Changes to paper:**
Whenever we wrote "our ensemble" in the text, we exchanged it to "sensitivity runs" or "model runs" as the number is too small to be named ensemble.

2. Page 4, sensitivity studies: The perturbations of soil moisture are applied to the relative water content w_so, irrespective of the soil type. Dependent on the soil type this results in a different perturbation. It would be better to first compute the soil-moisture index/soil moisture range/fraction of available water (theta-theta_wp)/(theta_fc-theta_wp), to then perturb this quantity, and to convert back to relative water content, as this would yield a fairer perturbation across soil types. Is the field capacity reached in the simulations with increased soil water content?

We take the different soil types into account **just by** perturbing the relative water content. The COSMO model uses the volumetric water content. E.g. to get spatial homogeneous soil water content, we need to adjust the volumetric water content in such a way that the relative water content is the same in the entire model domain. The relative water content (RWC) at each grid point is derived from the volumetric water content (VWC), the wilting point (WP) and porosity (PO), which are characteristics of the present soil type, in the following way:

$$\text{RWC} \quad = \quad \frac{\text{VWC} - \text{WP}}{\text{PO} - \text{WP}} \tag{1}$$

We included the formula and some comments on the calculation in section 2.2. The field capacity was not reached by changing the soil moisture in our model runs.

**Changes to paper:**

*"The relative water content is computed at each grid point from the volumetric water content (VWC) and the soil type dependent wilting point (WP) and porosity (PO) as follows:*

$$w_{so} \quad = \quad \frac{\text{VWC} - \text{WP}}{\text{PO} - \text{WP}}. \tag{2}$$

*"*

3. The smoothing of the orography not only modifies the static stability of the atmosphere, but also the moisture content by including/excluding parts of the moisture-laden lower troposphere. CIN/CAPE is one measure to predict the amount of convection that will occur. Another measure is the amount of moisture stored in the atmosphere that can be tapped by the convection. I suggest to include a plot showing the integrated precipitable water before the onset of convection for the different sensitivity experiments. On page 11, lines 11-12 it is stated that the total precipitable water changes. I think it is important to show these plots and to interpret the results. Page 1, line 20: "but that precipitation amount depends ..." in my opinion it is not only the strength of the trigger mechanism, but also the amount of moisture available for precipitation formation.

The remark in the text refers to modifications of the soil moisture which has a small impact on the precipitable water via evaporation. The largest changes (relative to the reference run) occur in the wet (SM_125) and dry (SM_075) experiments and range between -1.47% and +1.04%. As the reviewer correctly states, the smoothing of the terrain leads to a slight reduction of the terrain height. However, the response of precipitable water in those runs is an order of magnitude less than in the soil moisture experiments right before the convection is initiated (-0.61%–+0.28%). Due to theses small values, we decided not to include an extra figure, bur rather add one sentence in section 3.2 describing the orography perturbations:

**Changes to paper:**

*"We must further evaluate if the smoothing of terrain features, leading to somewhat lower terrain heights, has any implications on the precipitable water content. The analysis of the temporal evolution of precipitable water reveals that only marginal changes with respect to the reference run occur (relative deviations ranging between -0.61% and +0.28%), which indicates a negligible effect."*

4. Page 4, sensitivity studies: Please describe the generation of the random patterns in more detail.

For the generation of the random patterns we first use the python function numpy.random to create a random field consisting of 0 and 1. Subsequently, we convolve the pattern (using scipy.convolve). For this, we use two filters, which are derived from a Gaussian distribution, and which employ different expected values and variances. We then determine the length scale by looping through every 20th line in east-west direction and count the number of connected points with either positive or negative bias. We then average these values and find that the setting chosen for the small-scale random patterns has on average 20 connected points (10 km) with positive/negative bias and the larger-scale random patterns has on average 50 connected points.

We believe that such a detailed technical information does not need to be included in the paper, as the random patterns were generated just once and solely consist of dry and wet patterns at model initialization. Furthermore, the patterns are shown in Fig. 2. We slightly modified the text describing the random structures, it now reads:

**Changes to paper:**

*"Similarly, dry and wet patches were distributed randomly using a Gaussian filter, leading to*

*small-scale (SM_RS) or larger-scale (SM_RM) patterns."*

5. Page 5, lines 5-7: "we therefore conclude that ..." in my opinion this is not a valid statement as apples are compared with pears. The perturbation that is applied in the simulations with the altered soil-water content is much larger than just shifting around the soil water, but not changing its magnitude.

   There is no such statement on page 5. We believe that the reviewer refers to page 7 and sentence in the conclusions. We agree with the reviewer that changing the soil moisture has a much larger effect than just redistributing the reference soil moisture in uniform dry and wet patches. But that is exactly what we wanted to investigate and believe that this statement is valid. We changed this sentence to:

   **Changes to paper:**
   *"We therefore conclude that the initial soil moisture amount is much more important than the spatial distribution of dry and wet patches assuming a constant spatial average."*

6. Page 8, lines 23-24: "since the structure scales ..." what does this imply for the S values in the other sensitivity tests?

   For the CCN experiments, the structure component of the maritime runs are always lower than the remaining ones. This systematic behavior is not present for the other sensitivity. Because of that, we do not think that any implications for the other sensitivity tests need to be mentioned here.

   **Changes to paper:** none

7. Paragraph 3.2: within this paragraph an attempt to validate the SAL metric is mixed with results on the different perturbation techniques. This makes it hard to read the paper. The manuscript would benefit if you split the text into an evaluation part and a results part.

   When writing the first version of the paper, we also thought a lot about the structuring of the results. In the first part of section 3.2, only the 24-h averaged SAL-analysis is discussed. The analysis of the temporal evolution of the $A$-component is then mixed with other metrics to explain their differences. In our opinion, these points should not be separated. We already structured the text in the paragraphs Orography, Soil moisture, and Aerosols.

   **Changes to paper:**
   For a better distinction between the 24-h averaged analysis and the temporal evolution, we now moved the last point into a new section 2.3 called *"Factors determining the rain amount"*.

8. Page 9-10, section Orography: In the text the development of the grid-scale (resolved) convection is discussed. Yet, a parameterization scheme for shallow convection is active. When is sub-grid scale convection initiated in the different experiments, and how much of the instability is being eroded by the shallow convection parameterization scheme?

   From the existing code and model output, it is not possible to depict time and location of parameterized shallow convection. However, the parameterization of shallow convection is only used at 2.8 km grid length which serves as IBC for our 500-m runs. In the latter, the scheme is switched off as shallow convection is explicitly resolved at that scale. So, all sensitivity runs were run with 500-m grid spacing without any convection parameterization.

   **Changes to paper:**
   We added these remarks in section 2.1:

2.8 km:
*"Whereas deep convection is fully resolved, shallow convection is parameterized with a modified Tiedtke scheme with moisture-convergence closure (Tiedtke, 1989). Shallow convection is limited to a cloud depth of 250 hPa and is non-precipitating (see Baldauf et al. (2011) and Theunert and Seifert (2006) for details)."*

500 m:
*"Deep as well as shallow convection are now fully resolved and the Tiedtke schemes for shallow and deep convection are both switched off."*

9. How long are the simulations run for? Are there any spinup effects? Are the differences largest during the initial phase, or does the difference saturate, or even decrease, once the model has re-adjusted after the initial perturbation?

   All simulations had an integration time of 24 h. We included that information in section 2.1.

   In our COSMO simulations, spin-up effects are also present (e.g. increased wind-convergence or some weak and isolated precipitation). After 2–3 h of simulation time, no spin-up effects are apparent anymore, even for the strongest modifications (smoothing of the orography or large soil moisture gradients).

   **Changes to paper:**
   We also included that information in section 2.1.

Specific comments:

- Page 2, line 11: I suggest to replace "uncertainties" by "states" or "conditions", as no analysis of the underlying uncertainties is given.

  We replaced uncertainties with conditions.

- Page 3, line 1: replace "barrier" by "barriers", or include "a" before "stable"

  Done, the text now reads: barriers.

- Page 3, lines 14-22: to my knowledge there are two papers addressing the combined effect of soil-moisture perturbations and orography: Rihani et al. (2015), and Imamovic et al (2017).

  Thank you for pointing that out. Both of these papers use idealized simulations to isolate effects of terrain and soil moisture. We included them together with some new text:
  **Changes to paper:**
  *"Up to now, there exist only studies with idealized simulations on the isolated and collective effects of terrain and soil moisture heterogeneity. Rihani et al. (2015) conducted large eddy simulations and found that terrain effects dominate the planetary boundary layer development during early morning hours, while the soil moisture signature overcomes that of terrain during the afternoon. With convection-resolving simulations, Imamovic et al. (2017) found a consistently positive soil moisture-precipitation feedback for horizontally uniform perturbations, irrespective of the presence of low orography. However, a negative feedback emerged with localized perturbations. In both of these studies, terrain modifications were much more extensive via flattening of the idealized mountains. Moreover, uncertainties of the aerosol load were not addressed."*

- Page 4, line 1: replace "implies" by "includes"

  Done

- Page 4, line 13: replace "coefficents are active" by "diffusion is active"

Done

- Page 4, line 24: replace "physical" by "physically"

  Done

- Page 7, line 12: replace "dependancy" by "dependency"

  We changed dependance to dependence, there was no dependancy in this line.

- Page 7, line 13: include "a" before "heterogeneous"

  Done

- SAL technique: I suggest to use bold, italic or calligraphic letters for A, L, or S in the text.

  We now use italic letters throughout the entire manuscript, without marking it in the track-changed version.

- Page 8, line 10: replace "as" by "than"

  Done

- Page 11, line 9: replace "as" by "than"

  Done

- Page 11, Aerosols section: what about the collection processes of the falling cold hydrometeors, e.g. self-collection of graupel and hail, collection of rain by graupel or hail?

  The collection of rain or cloud water by cold hydrometeors is included in the riming rate, this was already stated in the text: *"...riming (collision of a droplet and an ice crystal)"*. The self-collection of graupel and hail is not included in the two-moment scheme of the COSMO model.

  **Changes to paper:** none

- Page 12, line 19-21: "we therefore conclude that...more important...", more important for what? As stated earlier, I dont think that this is a very meaningful statement.

  Please see our reply 5.:

  **Changes to paper:**
  *"We therefore conclude that the initial soil moisture amount is more important than the spatial distribution of dry and wet patches assuming a constant spatial average."*

- Page 12, line 29: either remove "to" before "precipitation", or replace "yield" by "lead"

  The text now reads: *"...can lead to precipitation deviations..."*

- Page 12, line 32: replace "forecast" by "forecasting"

  Done

- Figure 4: the symbols indicating the different sensitivity runs are very hard to read.

  We increased the size of the symbols for better readability. Since even bigger symbols will overlap each other, we cannot increase the symbol size any further.

- Figure 6: I suggest to include the reference simulation at (0,0)

The reference simulation is obviously positioned at (0,0). An extra point at that place would make Fig. 4 (e) and (d) harder to read, as many other points lie close to zero there. We therefore decided not to include the reference simulation.

Rihani, J. F., F. K. Chow, and R. M. Maxwell. 2015. Isolating effects of terrain and soil moisture heterogeneity on the atmospheric boundary layer: Idealized simulations to diagnose land-atmosphere feedbacks. Journal of Advances in Modeling Earth Systems 7, 915-937.

Imamovic, A., L. Schlemmer, and C. Schär. 2017. Collective impacts of orography and soil moisture on the soil moisture-precipitation feedback. Geophysical Research Letters, 44, doi:10.1002/2017GL075657.

**Reviewer 2**

General comments:

The results from simulation with varying soil moisture, orography (including other external data) and CCN concentrations were compared to each other and conclusions that soil moisture and CCN concentration can affect precipitation significantly were drawn. However, it is not clear to me, how the variation in the initial model conditions compare to each other. Soil moisture differences compare to differences which can occur between observed and modeled soil moisture content. CCN concentrations are described as varying between maritime and polluted conditions. How does this compare to observations? Modification in the orography are based on smoothing to coarser model resolution. How strong do these changes vary eg. compared to variation which can be achieved by tuning the orographic smoothing? Instead of addressing the question on how the resulting precipitation differs, I would rather ask the question how much one variable (eg. aerosol) needs to be changed to achieve the same model spread in precipitation as by a change in soil moisture content by eg. 25%. In the end the fact that soil moisture affects precipitation strongest can also be a result of the strength by which the soil moisture was modified. Preferable to do this an independent ensemble spread would provide a first estimate or the comparison to observational data which are present from the previous study from Schneider. Also possible to use the coarse scale simulation as reference comparison. While a lot of effort was put in conducting a very comprehensive model study, the analysis is lacking and often confusing. For example, mean values of the SAL score are calculated which are later revised as the A value is not representative. Page 10 Line 13 to 22 describe why the previously used values of A are not representative

To our knowledge, this is the first time that the relative contribution of three specific sources of uncertainty involved in the initiation and formation of convective precipitation is investigated. Further, the distinction of different weather regimes comprises a novel element highlighting their larger impact during weak synoptically forced weather, an effect often being veiled when using bulk forecast statistics.

The three types of perturbations were specifically chosen as they are involved in the initiation and formation of convective precipitation at different stages of the process chain. We agree with the reviewer that it is difficult to compare our three sensitivities, as the relative strength of modification is hard to establish. But our perturbations are chosen based on realistic values. The soil moisture difference of 25% was selected because Hauck et al. (2011) showed that simulated and observed soil moisture in southwestern Germany differ around 20–30%. The soil moisture perturbation length scales were also used in earlier work (Baur et al. 2018). The CCN concentrations was varied between maritime and polluted conditions, all of which may exist in the domain under investigation. For the terrain forcing, we wanted to apply a method for the entire model domain for comparability to the other sensitivities, instead of more radical terrain modifications, e.g. the flattening of individual mountain ranges (see Schneider et al. 2018) which locally can have a strong effect. The terrain modification seems to be the one with the least effect. On average, those modifications have the smallest impacts on the 24-h precipitation amount. However, on half of the days analyzed here, they have a similar range of impact than different aerosol assumptions. Therefore, we believe that our conclusions are still valid for our specific modifications.

Usually, the SAL-analysis is conducted for longer time periods as e.g. the 24-h precipitation amount. As the timing of precipitation is then not taken into account, we decided to compute SAL-values based on hourly precipitation as e.g. in the study of Henneberg et al. (2018). The averaged SAL diagram then gives a more reliable information as the SAL-values computed only from the 24-h precipitation amount. As positive and negative values can cancel themselves out during the period considered, we decided to also show the temporal evolution of the A-component. We believe that this is not a

weakness of our method, but rather a benefit of having both information.

Specific comments:

- Abstract: Page 1 Line 13: correct initial values are much more important than the spatial distribution... What is meant by correct initial values? I think what is meant is something like: The amount of soil moisture affects precipitation stronger than its spatial distribution.

  Yes, that's what we meant. We adapted the text accordingly.

  **Changes to paper:**
  *"We therefore conclude that the initial soil moisture amount is much more important than the spatial distribution of dry and wet patches assuming a constant spatial average."*

- Page 1 Line 18: ...that the structure component is highest in the soil moisture... Are structure values really higher for all soil moisture and CCN simulations or is structure most sensitive to changes in soil moisture (content) and CCN concentration. Also, I would avoid talking about structure in the abstract as the concept of SAL is not clear to the reader yet. What does structure mean? – Intensity of precipitation?

  On average, the highest S values occur for the aerosol and soil moisture modifications. The mean values were presented in section 3.2. We added *"on average"* and a short explanation for the structure component in the abstract which, to our opinion, should be sufficient information for the reader:

  **Changes to paper:**
  *"These diagnostics reveal that the structure component, comparing the size and shape of precipitating objects to the reference simulation, is on average highest..."*

- Page 1 Line 19: ...dominant mechanism for convection initiation... trigger mechanism What are these mechanisms?

  The trigger mechanism are e.g. low-level wind convergence with subsequent lifting of air parcels. This is analyzed in section 3.2. We do not think that a further explanation in the abstract is needed for that.

  **Changes to paper:** none

- Page 1 Line 20: Location and amplitude parameters are both much smaller. Change to: Location and amplitude parameters both vary over a much smaller range.

  Done

- Introduction: Missing explanation about the soil moisture precipitation feedback.

  There is an entire paragraph related to the soil moisture–precipitation feedback in the Introduction (P2, L23–P3, L4). We believe that it is sufficiently addressed.

  **Changes to paper:** none

  Missing: The conclusion of the study is to include soil moisture and aerosol variation in ensemble forecast to achieve a sufficient model spread? An overview of how ensemble spreads are generated in current ensemble setups would be beneficial.

  This is a good point, we included this text in the Introduction:
  **Changes to paper:**
  *"Ensemble forecasting has become a standard tool for probabilistic numerical weather prediction and most major meteorological services now run such systems routinely (e.g., Bouttier et al.*

*(2018)). Key uncertainties that are accounted for comprise, e.g., the uncertainties in the initial and lateral boundary conditions as well as uncertainties in the representation of physical processes (e.g., Clark et al. (2016), and references therein.)"*

- Page 2 Line 26: What is meant by soil moisture assumptions? Assumptions in the formulation in the couple soil model or assumptions about the soil moisture content?

  We mean assumptions about the soil moisture content and adapted the text accordingly.

- Page 3 Line 17: Aerosol or CCN? I think the study is constrained to changes in the CCN concentration while the freezing /nucleation scheme remains untouched?

  Aerosols serve as cloud condensation nuclei (CCN) and ice nuclei (IN), therefore we use both terms *"aerosol"* or *"CCN concentration"* in the text. In the model, we just change the CCN concentration and not the IN concentration. We included that information in section 2.1 describing the model set-up:
  **Changes to paper:**
  *"Heterogeneous ice nucleation on aerosol particles serving as ice nuclei (IN) is parameterized following Phillips et al. (2008) with the IN concentration left constant throughout the simulations."*

- Methods: Why chess board like structure to modify soil moisture field. That looks like a very artificial change and could cause artificial circulation. Is the domain average soil moisture content the same? How are the random perturbations generated?

  We agree with the reviewer that a chess board structure is a very artificial and idealized change. However, we believe that the uncertainty in soil moisture can be represented by such structures, as e.g. after scattered showers or after postfrontal convection, roughly similar wet and dry patches can be created. Chess board structures were also used by Courault at el. (2007) or Baur et al. (2018). Further studies with chessboard-type inhomogeneities of the surface are e.g.:

  Tsvang, L.R., M.M. Fedorov, B.A. Kader, S.L. Zubkovskii, T. Foken, S.H. Richter, J. Zeleny, 1991: Turbulent exchange over a surface with chessboard-type inhomogeneities. Bound.-Layer Meteor. 55, 141160.

  Friedrichs, K, Mölders, N, Tetzlaff, G (2000): On the influence of surface heterogeneity on the Bowen-ratio: A theoretical case study. Theor Appl Climatol 65: 181-196.

  Rieck, M., C. Hohenegger, and C.C. van Heerwaarden, 2014: The Influence of Land Surface Heterogeneities on Cloud Size Development. Mon. Wea. Rev., 142, 38303846, https://doi.org/10.1175/MWR-D-13-00354.1

  Lee, J.M., Y. Zhang, and S.A. Klein, 2019: The Effect of Land Surface Heterogeneity and Background Wind on Shallow Cumulus Clouds and the Transition to Deeper Convection. J. Atmos. Sci., 76, 401419, https://doi.org/10.1175/JAS-D-18-0196.1

  The generation of secondary circulations along soil moisture gradients has been documented in many studies and is an intended feature of our simulations. As we are aware that the chessboard structure is highly idealized, we additionally introduced randomly distributed dry and wet patches.

  For the generation of the random patterns we first use the python function numpy.random to create a random field consisting of 0 and 1. Subsequently, we convolve the pattern (using scipy.convolve). For this, we use two filters, which are derived from a Gaussian distribution, and which employ different expected values and variances. We then determine the length scale by

looping through every 20th line in east-west direction and count the number of connected points with either positive or negative bias. We then average these values and find that the setting chosen for the small-scale random patterns has on average 20 connected points (10 km) with positive/negative bias and the larger-scale random patterns has on average 50 connected points.

We believe that such a detailed technical information does not need to be included in the paper, as the random patterns were generated just once and solely consist of dry and wet patterns at model initialization. Furthermore, the patterns are shown in Fig. 2. We slightly modified the text describing the random structures, it now reads:

**Changes to paper:**
*"Similarly, dry and wet patches were distributed randomly using a Gaussian filter, leading to small-scale (SM_RS) or larger-scale (SM_RM) patterns."*

- How exactly are the inhomogeneous CCN concentration generated? Even though, CCN concentration can vary spatially I would assume they are advected similar to the clouds (at least in strong forcing cases). Clouds which travel through region with strongly varying aerosol conditions seem to be rather unrealistic to me and I fear this causes unphysical affects? In weak forcing cases I would assume the effect of this CCN perturbation is randomly depending on where convection is triggered with respect to the CCN modification.

Our model has 4 prescribed CCN concentrations and therefore we decided to use these values. We created squares with 112 x 112 grid points (56 x 56 km to resemble the soil moisture chess boards) that have the same CCN concentration and then placed 6x4 of these squares together (by hand) such that the mean concentration is close to 1700 cm-3 (as in the reference simulation), and that no squares with the same concentration are located next to each other, except that we allow for diagonal connections. These large patterns were then repeated throughout the domain (Fig. R.2). Note that CCN themselves are not transported in our simulations.

[Figure]

Figure R.2: Spatially heterogeneous CCN concentrations in the VAR experiment.

We believe that it is not unrealistic that clouds travel through regions with varying CCN concentrations, as e.g. larger cities or industrial regions can lead to local polluted conditions. Moreover, our patch size is comparatively large, so that convective clouds do not travel over many different patches (in particular for weak synoptic forcing). No artificial errors are introduced when using

the horizontally-varying CCN. In idealized tests, cells moving across a CCN boundary adapted slowly to their new CCN environment only through changes to the number of cloud drops activated (consistent with the new CCN environment). This means convective cells moving into regions of higher CCN concentration change relatively quickly (5-10 mins; as a larger number of cloud drops are activated near cloud base and are transported through the cloud), whereas cells moving into regions of lower CCN concentrations take longer to adjust (20-30 mins; because fewer cloud drops are activated near cloud base, but the large number of cloud drops existing throughout the cloud persist). This is exactly consistent with how a real convective cell would react to changes in the CCN environment.

For the weakly forced cases, we agree with the reviewer that aerosol–cloud interactions will depend on where convection is initiated. However, convection is widely distributed in our model domain (see Fig. 3 in manuscript), so that the clouds form in all four different CCN concentration patches (Fig. R.2).

**Changes to paper:** none

- I assume the changes in the smoothing of external data affecting the orography is only relevant in regions with complex topography, while the modifications in soil moisture and CCN concentrations are applied across the whole model domain.

  Yes, that is correct, the smoothing of the terrain primarily influences complex terrain. However, the central and in particular the southern parts of Germany are characterized by low mountain ranges which are affected by our modification technique.
  **Changes to paper:** none

- Page 4 Line 24: Change to homogenous soil moisture is done for all model levels. From that it is not clear if all soil moisture perturbations are applied on all soil layers.

  Yes, all soil model levels have been modified in the same way. We deleted the first remark and included this sentence at the end:
  **Changes to paper:**
  *"To assure physical meaningful soil moisture profiles, all soil moisture modifications mentioned above are done for all soil model levels."*

- Table 1: Maybe I missed it: what is the aerosol concentration in the reference run? In the text it says simulations with modified terrain and soil moisture run with 1700 cm-3. Is this also the case for the reference run? If so, I wonder what the difference to CON is.

  The reference simulation uses the continental aerosol assumption of 1700 cm$^{-3}$.
  **Changes to paper:**
  We inserted a sentence in section 2.1 to make that clear. There is no difference to CON, we therefore included some text in the heading of Table 1 and included CON(=REF) in the aerosol block of the table.

- The simulations were compared to radar observations in a previous study. The present study would highly benefit from including this comparison eg. also by using SAL analysis. This provides a reference deviation for a better quantification of the variations in precipitation that occur from different model settings.

  In the previous study of Schneider et al. (2018), only a qualitative comparison of Radar-derived with the simulated precipitation amount of the reference runs was made. As in this study, we believe that a qualitative good representation of the weather characteristics is sufficient and a quantitative analysis not necessary, as the focus lies on the sensitivity of the model to orography,

soil moisture, and CCN concentrations. Moreover, the radar-derived precipitation has a coarser resolution which would require an interpolation with loss of small-scale details. We therefore did not include any new figures or text.

**Changes to paper:** none

- Results: For the analysis SAL was used. While the A value mainly describes the changes in precipitation amount, it does not give much more information as the comparison of the precipitation amount. However, the S value gives information of whether precipitation becomes more intense and locally constrained (deeper convection) or increases in size. For that it is important to have A as comparison in order to derive if a change in structure is caused by a change in the area or intensity. This connection was not drawn in the analysis. Further, the averaging over SAL values over time diminish some effects. I'm not sure what the worth of an averaged SAL values are. Especially later the relevance and correctness of the A-value is often questioned.

  Usually, the SAL-analysis is conducted for longer time periods, e.g. the 24-h precipitation amount. As the timing of precipitation is then not taken into account, we decided to compute SAL-values based on hourly precipitation as e.g. in the study of Henneberg et al. (2018). The averaged SAL diagram then gives a more reliable information as the SAL-values computed only from the 24-h precipitation amount. As positive and negative values can cancel themselves out during the period considered, we decided to also show the temporal evolution of the A-component. We believe that this is not a weakness of our method, but rather a benefit of having both information.

  The connection between S and the rain intensity/amount was already described at in the section about the averaged SAL-values. We believe that a more detailed information (e.g. the correlation of S and A) is not absolutely mandatory for the reader to understand our main points.

  **Changes to paper:** none

- Page 6 Line 17: ...the sensitivity to terrain forcing always shows the smallest spread. As already mentioned above, I find this hard to judge as the sensitivity highly depends on the strength on variation.

  We agree with the reviewer that the terrain modification seems to be the one with the least effect. On average, those modifications have the smallest impacts on the 24-h precipitation amount. However, on half of the days analyzed here, they have a similar range of impact than different aerosol assumptions. There are more radical terrain modifications possible, e.g. the flattening of individual mountain ranges (see Schneider et al. 2018) which locally can have a strong effect. However, we decided to use the smoothing of terrain features as this is a modification done in the entire model domain and thus better comparable to soil moisture and CCN modifications.
  **Changes to paper:** none

- Page 6 Line 25: How does stratiform precipitation match the title?

  The cases under strong synoptic forcing do have stratiform and convective (embedded) precipitation. As the convective parts dominate the rain totals, we believe that our title is still correct.
  **Changes to paper:** none

- Page 8 Line 2-3: Change small to negative Change too small to smaller. Too smaller sound as if this is wrong but it is just different to the reference case. Change large to positive.

  Thanks for pointing that out, it is corrected.

- Page 8 Line 11: Are the SAL values smaller compared to other studies because of the model to model comparison or because of the averaging?

  In general, they are smaller because we compare the sensitivity runs to the reference run and not an observation. However, the averaging can also lead to smaller values as positive and negative values can cancel themselves out.

  **Changes to paper:** none

- Page 8 Line 29: On weak forcing days, there are simulations, in which the amplitude does not reflect the precipitation sum? So, what is the sense of the previous analysis than. This makes it really difficult to follow. Also on Page 10 Line 21.

  In this line, the days with strong synoptic forcing is discussed. In that section, we present the results from the averaged SAL-analysis which can be influenced by positive and negative values which can cancel themselves out and lead to a small A component. This is exactly why we added the temporal analysis of the A-component to better explain the precipitation totals.

  **Changes to paper:** none

- Page 10 Line 7: Change especially to only.

  Done

- Page 10 Line 24: Bowen ration not introduced yet. What does Bowen ration above 1 mean? Higher latent or sensible heat?

  Thanks for pointing that out. We included an explanation, the text now reads:
  **Changes to paper:**
  *"During daytime, the Bowen ratio $\beta$ (i.e. the ratio between the sensible and latent heat flux) increases to values larger than 1 in the SM_075 simulation as a result of the dominating sensible heat flux."*

- Page 10 Line 28: What is the relation between CAPE and A-component? Why does precipitation increase with reduced CAPE? (If my interpretation of positive A is correct). In Line 30 the argument is, that enhanced CAPE enhances precipitation. In this argumentation I miss arguments about the changed moistening of the atmosphere, what is needed to trigger, convection, destabilize the atmosphere and also to provide enough moisture for precipitation. Showing some more results about convective parameters such as surface temperature, CAPE or LCL developments may support the argumentation.

  Thanks for pointing that out, our first text version was misleading. We rewrote that part:
  **Changes to paper:**
  *"As a result of the weaker latent heat flux, the lifting condensation level is higher (not shown) and CAPE is reduced compared to the SM_125 simulation. Despite that reduction in CAPE, the model still simulates higher rain intensities in the SM_075 simulation than in the reference run. This can be explained by the stronger lifting from low-level wind convergence and the fact that there is still enough CAPE in the atmosphere to allow for deep convection to develop. This leads to higher rain intensities between 10:00–14:00 UTC compared to the reference or SM_125 simulation which are also represented by a positive A (Fig. 8i)."*

  For the sake of brevity, we decided not to show additional variables (T2m or LCL) as suggested, because we believe that the text alone is sufficient.

- Page 12 Line 1: What kind of precipitation is found below cloud base (rain or snow). Is it only a size argument what makes them less susceptible to evaporation (or sublimation)?

Below cloud base, the model simulates rain only (at least when averaged over the model domain) and no snow or graupel. The effect of evaporation is based on a size argument, but it was also evident from process rates and derived particle size distributions.
**Changes to paper:** none

- Page 12 Line 20: What are correct initial values?

  We rephrased that sentence as also reviewer 1 had concerns:
  **Changes to paper:**
  *"We therefore conclude that the initial soil moisture amount is more important than the spatial distribution of dry and wet patches assuming a constant spatial average."*

- Technical comments: Page 2 Line 10: the state of the atmospheric Atmosphere or atmospheric condition

  It should be *"state of the atmosphere"*, it is corrected.

- Page 11 Line 27: switch 28 July and 11 September as in the text above 28 July is also mentioned first

  Done

[revised manuscript text omitted]